# Insect-Inspired Robots: Bridging Biological and Artificial Systems

**DOI:** 10.3390/s21227609

**Published:** 2021-11-16

**Authors:** Poramate Manoonpong, Luca Patanè, Xiaofeng Xiong, Ilya Brodoline, Julien Dupeyroux, Stéphane Viollet, Paolo Arena, Julien R. Serres

**Affiliations:** 1Embodied Artificial Intelligence and Neurorobotics Laboratory, SDU Biorobotics, The Mærsk Mc-Kinney Møller Institute, University of Southern Denmark, 5230 Odense, Denmark; xizi@mmmi.sdu.dk; 2Bio-Inspired Robotics and Neural Engineering Laboratory, School of Information Science and Technology, Vidyasirimedhi Institute of Science and Technology, Rayong 21210, Thailand; 3Department of Engineering, University of Messina, 98100 Messina, Italy; 4Department of Biorobotics, Aix Marseille University, CNRS, ISM, CEDEX 07, 13284 Marseille, France; ilya.brodoline@univ-amu.fr (I.B.); stephane.viollet@univ-amu.fr (S.V.); 5Faculty of Aerospace Engineering, Delft University of Technology, 52600 Delft, The Netherlands; j.j.g.dupeyroux@tudelft.nl; 6Department of Electrical, Electronic and Computer Engineering, University of Catania, 95131 Catania, Italy

**Keywords:** hexapod, legged robotics, biomimicry, biomimetism, bionics, biorobotics

## Abstract

This review article aims to address common research questions in hexapod robotics. How can we build intelligent autonomous hexapod robots that can exploit their biomechanics, morphology, and computational systems, to achieve autonomy, adaptability, and energy efficiency comparable to small living creatures, such as insects? Are insects good models for building such intelligent hexapod robots because they are the only animals with six legs? This review article is divided into three main sections to address these questions, as well as to assist roboticists in identifying relevant and future directions in the field of hexapod robotics over the next decade. After an introduction in section (1), the sections will respectively cover the following three key areas: (2) biomechanics focused on the design of smart legs; (3) locomotion control; and (4) high-level cognition control. These interconnected and interdependent areas are all crucial to improving the level of performance of hexapod robotics in terms of energy efficiency, terrain adaptability, autonomy, and operational range. We will also discuss how the next generation of bioroboticists will be able to transfer knowledge from biology to robotics and vice versa.

## 1. Introduction

Legged robots represent a unique opportunity to understand locomotion in the animal kingdom [1,2]. Ever since *robotics* emerged in the 20th century, legged robots have aroused enthusiasm and curiosity from both researchers and the general public. In addition to their contributions to our understanding of locomotion, legged robots offer a great alternative to their wheeled counterparts, showing better abilities to navigate uneven terrains. Indeed, most animals have legs, allowing them to move, explore, and adapt to their direct environment. Similarly, legged robots can operate on many types of surfaces, including rough terrains, and also jump over obstacles, and climb structures. The following article is focused on hexapod robots, i.e., robots equipped with six actuated legs. The choice of hexapod design is strongly motivated by the need for insectoid robots to mimic the animal model as far as possible, so that the results obtained from experiments can be fairly compared to insects (locomotion, navigation, object manipulation, etc.) [3,4,5]. On the other hand, and from a purely robotic perspective, hexapod robots represent an optimum compromise between overall stability and energy cost. With fewer legs, such as in bipedal or quadrupedal robots, the locomotion pattern implies that whenever the robot is walking, its stability decreases during the leg transfer phase: a bipedal robot will have to stay stable with only one leg, and a quadrupedal robot is inherently unbalanced because of the symmetry of the legs configuration. Alternatively, octopod robots are stable, but less energy efficient. To summarize, hexapod robots offer different ranges of relatively stable walking gaits, while optimizing energy cost.

This review article was written to address a common research question in the field of hexapod robotics: *what is the best way to build autonomous hexapod robots that can exploit their biomechanics, morphology, and computational systems to achieve autonomy, adaptability, and energy efficiency comparable to a small living creature, such as an insect?* To help researchers to find relevant directions over the next decade in the field of hexapod robotics, we will divide this review in three main sections following the logic of Figure 1. These are: Section 2, “Biomechanics in hexapod robotics”, which will be focused on the design of smart legs; Section 3, “From biomechanics to locomotion”, which will be focused on insect and robot locomotion control,; and lastly Section 4, “From locomotion to cognition”, which will be focused on planning body actions to reach complex goals. Figure 1 illustrates the links between these three sections as they relate to the above main research question in italic, which is quite complex. To address it, it requires to divide it into more specific research questions about how we can transfer knowledge derived from biological agents to artificial agents. In addition, according to the scaling factor between a biological agent (i.e., an insect) and an artificial agent (i.e., a robot), the transferability of biologically-based knowledge will be discussed from a robotic point of view, because there are occasions when such transferabilities are unfortunately not relevant. As a result, we have deliberately limited the current state-of-the-art as seen in Table 1 to provide an overview of hexapod robots developed over the last 20 years in the range of 1–27 kg.

Section 2 aims to survey a significant quantity of studies reported in the literature produced over the two last decades in legs, structure, and morphology in hexapod robotics. The design of legged robots is a technical and technological challenge in many respects. Since the very first robots were built in the late 1960s [7,8,9,10,11], an outstanding body of work has been realized to improve both the mechatronic structure and the locomotion gait towards auto-adaptability to the terrain. Nowadays, after barely more than 50 years of research and development, the principles of robots are well mastered and leverage multiple opportunities in terms of field implementation (for a review of hexapod robots up to 2010s, see [12,13]). Over the past ten years, the increasing use of 3D printing has significantly boosted the development of brand new mechatronic designs for inexpensive hexapod robots, which are easy, for academics and students, to build, duplicate, and fix. This section will deal with passive materials, actuators, and force sensing in legs design to improve the level of autonomy of hexapod robots over complex terrain allowing future hexapod robots to maintain their speed over an extended period or range. Section 2 will be divided into the following three subsections:Section 2.1. Robotic leg design;Section 2.2. From legs to robots;Section 2.3. Hexapod robots’ accessibility criteria for academics.

Section 3 focuses on the considerable amount of research over the last decades on locomotion control in insects and hexapod robots. An insect’s central nervous system can efficiently combine information from a variety of sensor modalities to achieve interlimb and intralimb coordination, as well as control joint compliance for locomotion in complex terrains and object manipulation. Understanding such a control system and implementing it on a hexapod robot are still difficult tasks. To date, significant progress on the investigation of insect locomotion control and the development of hexapod locomotion control has been made. In this context, Section 3 provides the key findings of insect locomotion control and their translation to robot locomotion control (i.e., bio-inspired control). It also covers conventional engineering-based control and advanced machine learning-based control methods for hexapod locomotion generation. This is to update and guide our research community with future research directions in locomotion control. Section 3 will be divided into the following two subsections:Section 3.1. Insect locomotion control;Section 3.2. Robot locomotion control.

Section 4 explores a considerable amount of research produced over the last decades, which has concentrated on unraveling the main features of the insect brain and deriving the main guidelines for new approaches to high level behavior emergence in engineering. Insects will be presented as creatures able to show complex behaviors in very small brains. We then assume the possibility of the physical implementation of similar behaviors within a currently existing hardware device. Of course, to perform such a task, a deep level of knowledge about the loop linking the neural–functional–behavioral levels is needed. As with higher brains, complex behaviors in insects typically arise from the coordinated action of concurrent control systems, but the details of the neural mechanisms in charge of each function, and how these are coordinated, still remain largely unknown in a lot of cases. Therefore, Section 4 reviews some successful examples in which this task was performed, identifying, in reference to specific tasks, the responsible neural sites, the neuro-engineered models designed, and the final robotic structures showing the corresponding behaviors. Of course, a huge amount of work still remains to be done, especially as regards our capability to design suitable experimental setups leading to the explicit stimulation of other interesting behaviors. Section 4 will be divided into the following three subsections:Section 4.1. The fly brain and cognition;Section 4.2. The insect brain structure;Section 4.3. Insect brain functional models, implementations and robotic experiments.

Section 5 will deal with lessons learned from this review, and a conclusion will be drawn in Section 6 to give new lines of research and future directions in hexapod robotics for the next 10 years.

## 2. Biomechanics in Hexapod Robotics

Biomechanics are at the heart of the biomimetic embodiment of a hexapod robot (Figure 1a) because legs are constrained by mechanical stress when they are in contact with the ground, and they have to follow various locomotion patterns dictated by the locomotion control system (Figure 1b). Mechanical strength and execution fidelity of the pattern of locomotion by adding force sensing to joints are the two key parameters in biomechanics.

Below we introduce the recent developments in hexapod robot design reported over the last 20 years (Table 1), illustrating the outstanding achievements in size reduction, leg compliance, and speed in hexapod robotics. From a purely biomechanical point of view, three main research questions are explored in hexapod robotics:How can we improve the level of performance of hexapod robots in term of traveled distance?Is the study of insects helpful in the design of highly efficient hexapod mobility?Are robotic legs as intelligent as insect legs over complex terrain?

In Section 2.1, we introduce the variety of designs, morphology, actuation, cost of transport, and force sensing in robotic legs. In Section 2.2, we present the various hexapod robotic designs and the effect of the scaling factor on the level of performance. In Section 2.3, we show those hexapod robots, which are available on the market or as open-source projects for academics.

### 2.1. Robotic Leg Design

#### 2.1.1. Design and Morphology

The design of an insect-like hexapod robot is, in itself, a complex task, requiring compromises and the establishment of a balance between desired performance and true feasibility limited by technical progress. Two main aspects define a hexapod robot leg: the number of degrees of freedom (DOF) and the structure type. Each of these aspects are selected and designed for the uses to which the robot will be put.

**Number of degrees of freedom (DOF):** determines the operating space of the robot. By increasing of DOFs, the robot can achieve more complex trajectories. The number of DOFs also has a direct impact on robot characteristics, such as its autonomy, mass, and cost. Therefore, the number of DOFs should not be neglected in the design process. Currently, insect-inspired robotic legs are designed with between one and five DOFs per leg.With one DOF per leg, the robot’s maneuverability is highly limited. Depending on both leg and control designs, a one DOF per leg robot can perform a straight line walk [55] and also achieve simple rough terrain navigation if is equipped with *whegs* [46,56,57], comprising elements of both wheels and legs. These *whegs* equipped robots cannot really be considered as insect-like robots regarding their body structure. Their development tends to target navigation tasks over locomotion studies.With two DOFs per leg, a simplified hexapod robot can be built [58]. This choice is a good compromise between energetic cost and mobility. However, this type of robot walks mainly over flat terrains and can only perform curved leg trajectories, generating body oscillations.Previously, insect-based hexapod robots were built with three DOFs per leg (Table 1, in which the trochanter segment joint is merged with the femur and each joint only comprises one DOF, see Figure 2). Reflecting the standard insect leg model [59], this type of leg permits roaming in a slightly rough or slopped terrain in addition to a flat terrain walk.More DOFs in leg morphology improve maneuverability and adaptation to challenging terrains [39,60]. Additional actuators help to adjust robot orientation according to the slope in order to increase stability [26]. Experiments have shown that 4 or 5 joints per leg enable robots to cope with high gradient slopes in any orientation (e.g., up to $50^{\circ }$ slopes, see [26,61], or up to $43^{\circ }$ slopes, see [14]). Unfortunately, these improvements increase the level of complexity of control commands and the robot’s price and weight (Table 2), they also concomitantly, reduce autonomy due to the high power consumption of numerous actuators.To sum up, from the large number of robots based on three-DOF legs, this appears to be the right compromise to walk on a flat terrain. Despite the three DOFs per leg trend, from a biological point of view, an insect possesses more than three joints with one DOF per joint [62,63], allowing it to overcome large obstacles and cross sloped and rough terrain (e.g., up to $54^{\circ }$ see [64]). More complex models based more closely on insect leg kinematics are being developed [65]. Dung beetle like legs were built in 2018, the leg design was based on micro-CT scans of a real dung beetle [66]. A pair of beetle-like legs comprising four DOFs, allowing both manipulation and transportation was tested [66].In 2017, a hexapod robot, called Cassino Hexapod III (∼3 kg), composed of hybrid legs on a modular anthropomorphic architecture with omni-wheels, as feet at the extremities, was designed and built [67]. Each hybrid leg was built with three DOFs with the third being dedicated to rotating the wheel at the tip of the leg. This kind of hybrid locomotion is relevant for efficient rolling mobility on moderate terrain and walking mobility on extreme terrain, such as non-terrestrial exploration [68]. Hybrid locomotion by walking or by rolling allows hexapod robots to save energy, and this hybrid locomotion is a combination of an engineering approach and a bio-inspired approach. Hybrid locomotion has not been developed in this review, which is focused on the biomimetic approach.

**Structure type:** this describes how leg joints are linked to each other. Two major leg designs are used on inspect-inspired robots: serial multi-shaped legs or single shape legs.Serial multi-shaped legs are the most common structures encountered for locomotion and navigation. By definition, an insect leg is composed of five segments (coxa, trochanter, femur, tibia, and tarsus), arranged in a particular toggled zig-zag shape, forming a *sprawled* posture, reducing and distributing the forces on every joint of the leg [69,70]. However, in most robotic cases, this structure is simplified to three segments per leg (coxa, femur, tibia) comprising three joints per leg, each one with only one DOF (see previous point). In this arrangement, the trochanter segment of the leg is merged with the femur, and the tarsus is generally removed. However, the tarsus makes an important contribution to the insect’s walk, serving as an adhesive pad [62,71] and allowing a better ground forces transmission with a passive spring effect. Moreover, some insects (e.g., leafhoppers) possess particular tarsal structures allowing them to jump from smooth surfaces [72]. Looking over the last decade of hexapod robots, presented in Table 1, the tarsus is often neglected, even though it represents more than 30% of the leg length [73]. Currently, few artificial tarsus designs have been developed to efficiently walk on complex terrains [74].Single shape legs are used on *whegs* robots, origami, and compliant joint robots. The specifications of these types of robots, require the absence of most tiny mechanical parts such as bearings, shafts, screws, and nuts, and involves a simplicity of manufacturing, scale and cost reduction and, backlash and structural robustness improvements. The development of single shape legs follows the advances in new materials and manufacturing techniques such as multi-material 3D printing, which allows the building of soft joint robots [75]. Particularly, the 3D printing of legs appears to be a good way to develop and simplify standard joint designs by using properties of these new materials, such as flexibility or heat deformation [76]. In this way, hexapod robot legs are tending to become closer to real insect legs, in terms of relative dimensions and mass. An important point to notice for insects, e.g., cockroaches, is that a leg corresponds to approximately 2% of the body mass [77], allowing them low inertia, high frequency strides during a walk. In comparison, insect-like robot legs represent at least 10% of the overall mass (estimated for a 2 kg robot, from Table 1). Apart from 3D printing, some materials could take over from standard aluminum or molded plastic legs, e.g., chitosan–fibroin material, inspired by insect cuticle structure [78].Furthermore, some other original structures have been designed; they were mainly developed when a specific animal behavior, such as jumping [79,80], is to be replicated or to satisfy some sought after design specifications like posture changes [81].At first glance, leg design is highly dependent on actuator technologies. However, an impressive number of improvements are still possible through subtle structural modifications, allowing huge performance improvements. Independently of the structure type, observing the current state of the art in leg design, a question presents itself: why are all the legs of a hexapod robots the same? Insect legs are different in size (Figure 3, see [82]), and not built like robot legs, wherein the six are often identical, except for a few robots mimicking insect morphology in detail (Drosophibot [18,19] and MantisBot [30,31]). In response to this question and with the technologies now available, in the 2020s, leg design is likely to become increasingly based on available micro-CT scans of real insects (e.g., [66,83] dung beetles, [84] flies, or [65] ants) in order to improve the level of complexity, fidelity, and bio-inspiration.

#### 2.1.2. Cost of Transport

To compare the level of performance between robotic designs, the cost of transport (CoT) is now a common adimensional metric (Equation (Equation 1)) used to evaluate any legged robots [14,85,86,87].
(1)$$CoT=\frac{\bar{P}}{m\cdot g\cdot \bar{v}}$$
where $\bar{P}$ is the mean power consumption, *m* is the robot weight, $\bar{v}$ is the mean speed, and *g* is the gravitational acceleration (*g* = 9.81 m·s^−2^). The CoT depends linearly on the weight in log-log space with a negative slope [88,89,90] for both animals and robots. In the animal kingdom, biologists evaluate the efficiency locomotion with the mass-specific energy per unit distance (in J/(m·kg)), which can be defined by the gravitational acceleration-CoT product. With such an approach, we can estimate the ant’s CoT ∼39, and also the CoT of a 2–3 kg animal as close to ∼2. The current CoTs of hexapod robots weighting 2–3 kg is similar those of animals (Table 2). As a result, we can conclude that the locomotion efficiency of hexapod robots is currently no better than that of animals.

The CoT of hexapod robots is lower in tripod locomotion on a flat terrain [14]. A tetrapod or pentapod locomotion deteriorates the CoT value, likewise a sloping or rough terrain. As a result, to compare hexapod platforms, they must be evaluated using the same task and environmental conditions, here in tripod locomotion on a flat terrain and in a room with the air temperature set at 25 °C (Table 2).

#### 2.1.3. Actuation of the Legs

Many actuator technologies can be used to control the joints between the leg segments: servomotors, brushless motors, or artificial muscles, which are under developments in laboratories.

**Servomotors**: the main issues with servomotors are their weight and energy efficiency. A servomotor heats up easily until it surpasses its maximum operating temperature of 70 °C, then it stops working. In addition, a servomotor is composed by definition, of a motor with high ratio gearing used to make it as stiff as possible. In this sense, such servo-based actuators differ significantly from biological actuators that may have variable stiffness and adaptable compliance. One way to implement variable stiffness is to use springs to make variable impedance actuators (VIA). As summarized in [91], VIA actuators can be classified into three categories: spring pre-loaded variation, transmission ratio changing and spring physical property alteration. VIA is certainly an approach of great interest for the design of future hexapod robots able to dynamically change the stiffness of their joints.**Brushless motors**: recent developments in smart rotating actuators based on brushless motors will permit the design of direct drive joints without gearing. The maximum specific power of electric motors with permanent magnets is 300 W/kg, which is about the same order as biological muscle [92]. Companies, such as HEBI Robotics [17] or IQ Motion, have developed integrated rotating actuators for robotic applications and for the development of mobile robots of various sizes. As the electronic driver and angular sensor are integrated into the motor, it drastically simplifies the wiring and complexity of the overall hardware architecture, which can be crucial when designing robots like hexapods that require the control of 18 actuators.**Artificial muscles**: the design of future insect-inspired robots will certainly depend on the availability of actuators able to mimic the functionalities of biological muscles. Their properties of viscoelasticity and energy dissipation leading to high compliance is the holy grail of insect-inspired actuators. Among the broad repertoire of new artificial actuators for robots (see review by [93]), non-conventional actuators like pneumatic artificial muscles (PAMs), shape memory alloys (SMAs), and electroactive polymers (EAPs) are of great interest. One particular case is HASEL actuators, which are composed of a series of pouches made of a flexible and inextensible shell that is filled with a liquid dielectric. Electrodes cover a portion of each pouch so as to progressively close when a voltage is applied thus squeezing the pouches to increase their volume [94]. HASEL actuators can be implemented in different ways and can feature a bandwidth as high as 126 Hz for the quadrant-donut HASEL and even a specific energy twice as high as mammalian skeletal muscles for the planar HASEL actuator [94]. HASEL actuators mimic the muscle-like performance of dielectric actuators (DEAs), which can be highly effective for robotic applications. They can lift more than 200 times their weight and have a peak specific power of 585 W/kg [95]. Moreover, it is worth noting that a toolkit has been developed to aid designs using HASEL actuators [96]. In addition, electro-ribbon technology, with an ability of lifting 1000 times its own weight and a contraction by 99.8% of its length, is also very promising [97]. Finally, five-DOF soft dielectric elastomer actuators have been shown to be very useful in the implementation of soft legged robots [98] which are able to walk with an alternative tripod gait as fast as 52 mm/s for a 7 Hz actuation frequency.

#### 2.1.4. Force Sensing in Robotic Legs

An overview of the state-of-the-art in force sensing for multi-legged walking robots is available in this study [99], but where is the right location in a leg for sensing a mechanical action: is it at the tip of the leg, within the leg structure, or directly in the joints?

**Leg tip/TARSE sensing:** sensing at the end of the leg can be done by a tactile sensor, a pressure sensor, a three-axis force/torque sensor [100], or a compliant force sensor made with a spring [15,39]. Leg tip sensing can be easily implemented by adding an attachment point to the leg tip without requiring any modification of the robot’s structure. The cost of these tip sensors can be expensive depending on the chosen technology, but recent research has developed low-cost designs [100]. Tip force sensing is useful because it provides the robot with a terrain description. Force measurements allow the robot to understand which of its legs are in contact with the ground, or to evaluate the terrain slope, in order to both adjust its gait and plan its path [101].**Force sensing in actuators**: sensing coming from the state of actuators by current measurement [26,29,37,43] or dedicated sensors in the joints. This category of sensors simplifies the robot’s design, since the sensor is integrated within the actuator forming a compact structure. The complexity of the estimation of forces from the currents generated by the actuators is based on the robot’s leg model. To obtain an accurate leg movement, the mathematical model should reflect the robot as closely as possible, and take account of any structural deformation under various loads since no material is perfectly rigid.**Legs with compliant structures:** compliant mechanisms exploit the deformation properties of the leg segments, deformations that could be a disadvantage in other legs. Stiffer legs appear to narrow the region of stable gaits while preventing tripod contact with the ground. However, compliant legs are more capable of absorbing energy even if the leg touches down early, thus minimizing the severity of ground reaction on legs. This solution had been developed for one-joint C-shaped legs [44,45,54]. *Whegs* do not possess any force sensors on their legs. Compliant legs offer the possibility of placing the force sensors along the segments (such as the femur or the tibia) [101,102]. This type of sensor placement mimics the force measurements in insects, as done by *campaniform sensilla* mechanoreceptors [103,104].

Currently, three distinct groups of sensors exist and are implemented on board hexapod robots. A force sensing in actuators may be chosen if no structural modification is allowed. Alternatively, with a slight modification of the tarse, leg tip sensors provide the required information for achieving accurate locomotion control over rough terrains. Lastly, compliant legs permit the implementation of force sensing although this will increase the complexity of the design.

### 2.2. From Legs to Robots

#### 2.2.1. Body Morphology

Although hexapod robot leg morphology seems to be a similar, albeit simplified, representation of insect morphology, the overall body structure of most of the common robots is far from that of animals. Firstly, few robot designs feature a multi-segment thorax. While this is present in robots such as HECTOR [34,35,36] or MantisBot [30,31], the benefit of this type of design has still not been demonstrated. This only appears to help during tight turns or high obstacle climbing [105]. Thus, the increase in weight and energy consumption of this type of design, due to additional joints and body parts, justifies the more general choice of single shape body morphology. The second point is another aspect of body geometry. Various geometries are used in hexapod robot developments: circular, hexagonal, rectangular, etc. The main difference between these geometries is the necessity of performing a specific turning gait for the rectangular shape, whereas circular or hexagonal bodies allow, through their symmetry, an omnidirectional walk (see the review [13] for more details).

#### 2.2.2. Scale Effect on Level of Performance

Insect-like robot design process aims to validate biological hypotheses in navigation or locomotion, or to mimic behaviors and sensory systems (as shown in Table 1).The scale should then be taken into account throughout the development process as this will affect kinematic changes in robot locomotion and mechanics material [106]. Therefore, the question of the scale’s influence is raised. How will the scale affect the performance level? If a behavior is present in a small insect, is it possible to reproduce it at a given scale of robot? To answer these questions, it is necessary to discuss the kinematic changes in robot locomotion taking account of material mechanics and energetic aspects.

From the point of view of the material, the larger the robot is, the larger are going to be the internal deformations of its segments. To compensate for deformations, larger beams have to be used, so the weight of the leg segments and of the thorax are increased. Moreover, the scale factor of the robot proportionally affects the step size. However, not all kinematic parameters vary linearly with scale. From mechanical equations, some scale factors have been set [107,108], giving a scaling effect on essential physical values like mass, frequency, stiffness, damping, velocity, and power. Thus, a small size robot with a scale close to that of an insect can possess a light structure, fine legs, and walk at a high frequency. Conversely, for a large scale robot, higher mass and a slower frequency walk, due to inertia increase, are going to be observed. Higher actuator power will also be required, to set the robot in motion, and this implies a manufacturing price increase.

As a brief conclusion, the development of large scale hexapod robots is not justified. The advantages of working with a high scaling factor are their improvements in energetic values and technological limits (see Table 2). Regarding power consumption, represented by the CoT (see Section 2.1.2), the relationship between different scales of the same robot morphology has not yet been precisely fixed [89,108]. Basically, the global trend shows that, the more massive the hexapod, the lower its CoT; thus, it has a better walk efficiency. However, and by definition, the CoT does not take account of animal scale and its interaction with environment. The high CoT in insects must be considered in terms of their lifestyle, they do not need to save more energy when traveling. Accordingly, a robot’s size should be selected according to the type of mission (environment relief, distance to travel, payload, etc.), then corresponding to a given CoT value, roboticists should be able to compute the ideal robot mass.

Moreover, there are difficulties in manufacturing reduced scale robots. For instance, common methods of 3D printing have an average deviation error of 0.4 ± 0.2 mm [109], which limits the size of printed mechanics. Batteries also have technological limits. Commonly used lithium batteries have a low energy density (around 150 Wh/kg) [110]. Moreover, battery cells have a high mass, difficult for a light robot. In the near future, the development of new bio-inspired power sources should bridge this energetic gap, with a higher energy density (298 Wh/kg [110]), and this would allow more compact robot designs [111].

### 2.3. Hexapod Robot Accessibility Criteria for Academics

Over the past 10 years, many new designs have been proposed to the general public. These hexapod robots are generally available in *do-it-yourself* kit form, thus promoting opportunities for customization such as modifications in actuation and the addition of sensors to enhance the robots’ locomotion skills. For instance, the small inexpensive hexapod robot Hexy, produced by ArcBotics, features 18 DOFs achieved by means of Fitec servomotors (0.12 s per $60^{\circ }$), and is mainly intended for educational applications (Figure 4a). A leader in the market of robot kits for the general public, Lynxmotion™offers a wide range of relatively small designs, such as the Phoenix, which features 18 DOFs (Hitec HS-645 servomotors) with a maximum speed of 25 cm/s (Figure 4b), or T-Hex for which the 24 DOFs allow more complex motion (Figure 4d). Recently, EZ-Robot released Six, a fully 3D-printed hexapod robot endowed with 12 DOFs (Figure 4e). As with Hexy, this robot was released with educational purposes in mind, allowing students to tackle complex tasks such as visual-based object tracking and artificial intelligence with machine learning applied to gait generation. The 18-DOF PhantomX AX Metal Mark III hexapod robot (Figure 4c), developed by Interbotix Labs., offers promising dynamic performance with a maximum speed measured at 80 cm/s. Generally, these robots could be considered as inexpensive, with prices ranging from 250 to 3000 US dollars, but others require a significant budget, such as the Daisy Hexapod Robot Kit from HEBI Robotics [17], available from 83k US dollars (Figure 4f). The variations in price are almost exclusively the result of the quality and number of servomotors used in their design. On average, hexapod robots equipped with servos using only plastic gears are cheaper, but have a significantly reduced lifespan compared to their metal counterparts. Although standard servomotors are not optimized for such applications, they offer a quick solution for benchmarking new designs of hexapod robots in academia. Open-source projects by private individuals have also been reported. For instance, the MX Phoenix hexapod robot [27] and the MorpHex double hexapod robot were developed by Kare Halvorsen, who was inspired by robotic projects from Jim Frye (founder of Lynxmotion) and from Matt Dentons (Mantis project manager and founder of Micro Magic Systems). MorpHex’s unusual design features two hexapod robots connected in a top to tail fashion. Their legs are covered with spherical parts in such a way that when the robot refolds, it forms a perfect sphere. This mechatronic design allows MorpHex to either crawl or roll on almost any kind of terrain.

The aforementioned robots represent a unique set of opportunities for academia in both research and education since they are relatively inexpensive and easy to obtain, either in kit form or by replication based on open-source models. Fast customization for adaptation to the experimental context is another important feature of these new products. This recent trade has been intensified with the emergence of online sharing platforms such as *GitHub*, *Thingiverse*, and *Instructables*. A wide range of open-source projects can be found on these very popular platforms, supported by an extensive international community of robotics designers, engineers, and researchers. The recent democratization of 3D printing is also greatly contributing to the rise in open-source projects. Lastly, the development of a unique framework for robot programming, i.e., robot operating system (ROS), ensures the standardization of software developments, such as locomotion firmware for hexapod gait generation.

## 3. From Biomechanics to Locomotion

In the previous sections of this article, the relevance of biomechanical aspects has been mainly summarized in relation to actuation, robot leg, and morphology design (see Section 2). Supplementing biomechanical developments, locomotion control (Figure 1) is an important ingredient for interlimb and intralimb coordination as well as joint compliance (Figure 5). Interlimb coordination concerns the relationship between the legs for generating gaits, while intralimb coordination deals with the relationship between the joints within one leg for generating swing and stance, as well as adapting leg movements to deal with complex terrain [112]. Joint compliance can be beneficial for impact force absorption, payload compensation, disturbance rejection, and energy efficient locomotion [113,114].

In Section 3.1, we will introduce insect locomotion control for interlimb and intralimb coordination and joint compliance control. In Section 3.2, we will present different robot locomotion control methods including bio-inspired, engineering-based, and machine learning-based control. Finally, in Section 5.2, we will suggest some future directions in locomotion control for hexapod robotics.

### 3.1. Insect Locomotion Control

Insect locomotion control is based on neural mechanisms with a distributed control architecture [126,127] located in the insect’s thoracic ganglia. It is known that there are special neural circuits, called central pattern generators (CPGs) [128,129], with spiking and non-spiking interneurons for generating basic rhythmic motor activities. Activity can still continue even in the simultaneous absence of both afferent feedback and rhythmic inputs from other neural circuits. Biological investigations reveal that each hemisegment has independent CPG modules that regulate the motor neurons and muscles of the leg joints. As described in [129], CPG non-spiking interneurons (NSI E4, NSI I4, and NSI5) contribute to the generation of various leg movements. They can reset the phase and influence the frequency of the rhythmic neuron activity in leg motor neurons [130]. Specifically, NSI E4 was observed to deactivate the stance phase and activate the swing phase while NSI I4 was observed to induce searching movements [131].

Although CPGs are the basis of leg movement generation, sensory feedback is important for interlimb coordination and adaptation. For instance, joint movement signals are used to coordinate joints and switch leg movement states from swing to stance and vice versa. Furthermore, a combination of joint movement signals [132] and load signals [133] can entrain the CPG. This is known as entrainment. Studies on stick insects and cockroaches have identified interlimb coordination where neural signals from a front leg can entrain the CPG controlling the middle leg. Sensory feedback to the CPG and coordination for gait transition can occur at two levels: one is direct sensory input to the central descending command of the CPG (high-level feedback) [134] and another is local sensory feedback or reflexes which influence the CPG phase (low-level feedback) [135]. Typically, for interlimb coordination, it seems that slow-walking insects (e.g., stick insects) rely more on sensory feedback [136] than on the central coupling between CPGs, while fast-walking insects (e.g., cockroaches) rely more on centrally coupled CPG coordination (for more details see [129]).

In addition to CPGs and sensory feedback, some biological investigations have shown that forward models [137] also play an important role in insect locomotion control. The forward model is the neural mechanism that transforms motor commands (efference copies) into an expected sensory input in order to compare it to the actual incoming sensory feedback. It is used for state estimation or sensory prediction. This information can allow for movement adaptation. In the stick insects *Aretaon asperrimus*, when ground contact is lost at the end of the swing phase while climbing over extremely wide gaps, they immediately alter their leg stepping pattern [138]. This would indicate that ground contact is expected on a frequent basis. Other findings that support the concept of forward model predictions [116] show that stick insect responses to barriers are influenced by an internal state during the swing phase. In locusts [139] and fruit flies [140], they seem to have, when tested in flight simulators, forward models that can predict changes in their sensory feedback and adaptively modify the motor control parameters to cope with these changes. Current animal studies (mentioned above) indicate that CPGs, multimodal sensory feedback, and forward models are used in insect locomotion control for interlimb and intralimb coordination, movement adaptation, and joint compliance (all described below). The contributions of these components, however, vary across species.

**Interlimb coordination:** biological studies have revealed rules for interlimb coordination of insect locomotion. For instance, Wilson [119] proposed five rules. Rule 1: a wave of swing runs from hind (posterior) to front (anterior) legs. Rule 2: contralateral legs of the same segment alternate in phase. Rule 3: protraction (swing) time is constant. Rule 4: frequency varies (stance decreases as frequency increases). Rule 5: the intervals between steps of the hind leg and middle leg and between the middle leg and fore leg are constant, while the interval between the foreleg and hind leg steps varies inversely with frequency. These rules have been translated to neural mechanisms for hexapod locomotion control, which can generate various insect-like gaits [141,142].Subsequent research by Cruse et al. [116] introduced six rules for insect walking (called WalkNet). The rules were derived from behavioral experiments with stick insects. Rule 1: posterior swing inhibits start of anterior swing. Rule 2: start of posterior stance excites anterior swing (posterior reaches a given anterior extreme position (AEP)). The AEP is the anterior transition point from swing to stance in a forward walking animal. Rule 3: caudal positions of anterior stance excite start of posterior swing (anterior reaches a given posterior extreme position (PEP)). The PEP is the posterior transition point from stance to swing. Rule 4: end position of anterior stance influences end position of posterior swing (called targeting). Rule 5: increased resistance increases force and increased load prolongs stance phase. Rule 6: the information from the anterior leg’s reflex stimulation is passed on to the posterior leg. Recently, Schilling and Cruse [143] introduced the realization of these rules as an artificial neuronal network with an antagonistic structure (called neuroWalknet controller). The controller can generate diverse robot walking behaviors including different gait patterns emerging from different velocities, curve negotiation, and backward walking.In addition to the aforementioned rules for insect locomotion, a recent study from Leung et al. [144] analyzed and identified four underlying rules for interlimb coordination of dung beetle ball rolling gaits. Rule 1: front legs alternately step on the ground. The rule describes the relationship between the two front legs in the gait. Rule 2: each middle leg steps similarly to its contralateral hind leg. The rule describes the synchronization of the contralateral middle and hind legs. Rule 3: an ipsilateral pair of middle and hind legs seldom lift together. Rule 4: a contralateral pair of middle or hind legs rarely lift together. In principle, a pair of legs following the third and fourth rules tend not to lift together. A partial implementation of the rules as modular neural control with a CPG was performed and tested on a simulated dung beetle-like robot [4]. The controller can generate four different robot behaviors including forward walking, backward walking, level-ground ball rolling, and sloped-ground ball rolling.

**Intralimb coordination:** In addition to the biological studies of the relationship between legs (interlimb coordination) in insects, some studies have further investigated individual leg movements and adaptations during normal and rough terrain walking in insects. The leg movements basically reflect intralimb coordination. For instance, Pearson and Franklin [117] proposed locusts’ reflex strategies for leg movements when walking over rough and complex terrain. As described by them, the strategies include (1) rhythmic searching movements; (2) local searching movements; and (3) elevator reflex.Based on [117], the rhythmic searching movements are to search for a ground contact, if the animal has not located it by the end of its swing phase. The searching movements show rhythmic patterns including fast elevation and depression movements of the leg. While searching, the animal also extends the search range from the body to explore the supporting points around the leg, e.g., up to eight searching cycles. The searching typically stops either when the animal stops walking, the leg gets stuck, or ground contact is found. The local searching movements are small rhythmic leg movements from point to point on a potential supporting ground. These movements occur either at the beginning of a stance phase, after the rhythmic searching movements and/or an elevator reflex (described below). The local searching movements are required if the potential support surface is smooth where the leg action needs to push the animal forward. The elevator reflex is a rapid elevation and extension of the leg to step over an obstacle, followed by placing the leg where the obstacle can be used as a support. The elevator reflex can be activated when the leg gets stuck during the swing phase. It can also occur during searching movements. In rough terrain walking experiments on locusts, the elevator reflex was mostly observed in the fore and middle legs while the hind legs moved behind the animal. This makes it difficult to distinguish between an elevator reflex and a passive pulling of the hind legs up onto the obstacle while moving forward. Examples of the implementation of these reflex strategies for adaptive hexapod robot locomotion on rough and complex terrain can be seen at [118,142].

**Joint (mechanical) compliance:** this is the interactive relationship between kinematic changes and the resulting dynamics of joints [145]. One of its key components, stiffness, refers to the ratio between joint torque and angle changes, which are related to muscle activation. Compliance control of insect muscles is very important in facilitating adaptive and robust locomotion over natural terrain [146,147]. Computerized (computational) muscle models can be used to enhance the understanding of neuromechanical control principles underlying insect locomotion [148]. The Hill muscle is one of the most influential ‘seed’ models that has inspired many successors [149]. For instance, Proctor and Holmes built a neuromechanical model to study feedback effects of perturbated insect locomotion [150], in which 24 neural oscillators and 48 pairs of Hill muscles were used. Guo et al. proposed a neuro–musculo–skeletal model to reproduce gait pattern in virtual insects [151]. Naris et al. analyzed a closed-loop neuromechanical simulation of insect joint control driven by a pair of Hill muscles [152]. However, most of these studies were limited to numerical simulations, because a greater number of parameters needed to be offline optimized based on nonlinear differential equations. Therefore, they failed to account for intrinsically delayed feedback in real insect locomotion dynamics. This failure may cause a misinterpretation of the neuromechanical control principles in insect locomotion. Therefore, questions remain open whose answers may decode insect muscle intelligence in dynamic robust locomotion.-**Multifunctional muscles:** insects exhibit different muscle functions in flying and walking [153], which are characterized by the work loop technique [154]. These functions may facilitate the decoding of muscle compliance in dynamic insect locomotion. Interestingly, some preliminary results show that muscles act as brakes and springs when their passive stiffness and damping are tuned in computational simulations [155]. Tuning muscle stiffness and damping properties based on the work-loop technique, can be a key to understanding and translating muscle intelligence between engineering applications and neuromechanical models [115,156].-**Predictive muscle tuning:** muscle compliance can be tuned in terms of sensory feedback. However, this feedback is intrinsically subject to noise and delays owing to high levels of dynamics of insect locomotion. Therefore, it may be assumed that insects, and their robot counterparts use internal models to predict sensory outputs for tuning insect muscle compliance [114].-**Engineering-inspired muscle intelligence:** biological muscle control principles have been borrowed to enhance robot designs and control for many years. This research approach can be flipped, i.e., robots as tools for decoding muscle compliance in insect locomotion [157]. For instance, an insect-like robot was used to test a simplified muscle control hypothesis, i.e., proximodistal gradient [114]. It showed that this gradient reduces the number of controlled variables and enhances walking stability. Engineering-inspired methods can close the research loop of insect muscle intelligence, providing new hypotheses for biological experiments on insect locomotion.

### 3.2. Robot Locomotion Control

While insect locomotion control (Figure 1b and Figure 5) has been studied and thoroughly investigated as described above, translation from biological investigation to robot implementation remains a challenge. To date, different locomotion control methods for interlimb and intralimb coordination have been proposed (Figure 1b and Figure 5). The key methods from different domains include (i) bio-inspired control (e.g., pure CPG(s), pure reflexes, or their combination) (Figure 6a); (ii) engineering-based control (e.g., kinematic and dynamic models) (Figure 6b); (iii) machine learning-based control (Figure 6c); and (iv) a combination of these key methods. This review will focus on bio-inspired control while the other control techniques will be briefly discussed.

#### 3.2.1. Bio-Inspired Control

One of the standard bio-inspired methods is CPG-based control where CPG rhythmic signals are used to generate individual leg movements (intralimb coordination) and create coordination between the legs (interlimb coordination) [167] (Figure 6a, upper and lower insets). A number of CPGs have been explored for locomotion control from minimal to maximal CPGs (i.e., all legs and joints controlled by a minimal single CPG [168], each leg by one CPG [169,170] (six CPGs in total), or maximally each joint by one CPG [171] (18 CPGs in total)). Various CPG models have been developed (Figure 7) from conceptual biological CPG models based on a half-center oscillator [172], biophysical models using Hodgkin–Huxley neurons [173], connectionist models using simplified neurons with various activation functions [174,175,176] to abstract models using nonlinear coupled oscillators. Note that here CPG(s) and oscillator(s) are used interchangeably through out this section.

In principle, the CPG acts as an open loop control since it does not require sensory feedback to generate its periodic patterns. Most of the CPG models can generate only periodic patterns. To achieve a variety of complex patterns, including periodic and chaotic ones for complex robot locomotion, Steingrube et al. [141] modified the SO2 oscillator to become a chaotic CPG and applied a novel adaptive chaos control that exploits neural dynamics embedded in the chaotic CPG and uses time delay feedback mechanisms to control the dynamics. Although none of these abstract models require any external input or sensory feedback to produce basic rhythmic activity, they do need sensory feedback to adapt and tune their frequency, phase, and magnitude for efficient locomotion control to deal with different situations. Thus, different feedback techniques have been developed for frequency, shape, and phase adaptations.

For the frequency adaptation, typically, joint angle feedback and foot contact feedback are used to entrain the frequency of an oscillator (entrainment [186]). If the oscillator gets entrained by the feedback and adapts to it, even if only temporarily, i.e., the feedback only has a short term effect, such an oscillator is considered as reactive [177,179]. In this case, if the feedback is switched off the CPG system immediately returns to its intrinsic dynamics, there is no memory of the input and no lasting change to the dynamics. To maintain the effect of the feedback after the feedback has been removed, Righetti et al. [180] introduced a frequency adaptation schema, which permanently modifies the intrinsic frequency of an oscillator. An oscillator with this schema is considered to be an adaptive frequency oscillator (AFO). However, AFOs can suffer from significantly long adaptation times as has been shown in many robotic applications. To obtain fast as well as precise adaptations, Nachstedt et al. [183] proposed frequency adaptation through fast dynamical coupling (AFDC). It is based on dynamically adapting the coupling strength of sensory feedback to an oscillator. While the AFO and AFDC can automatically adapt the oscillator frequency to match the frequency of an external periodic signal (i.e., robot joint angle feedback, which can identify robot locomotion), they do not deal with the tracking error that may occur between the actual robot’s motion and oscillator output. This could lead to the loss of precision, unwanted movement, or energy-inefficient locomotion. To address this, recently Thor and Manoonpong [184] proposed online error-based learning for frequency adaptation of oscillators. The learning mechanism can reduce tracking and steady-state errors as well as perform fast and stable learning. Frequency adaptation is basically used for robot locomotion enhancement [16].

For the shape adaptation, a typical technique is to use additional premotor neuron networks for shaping oscillator signals [49,187]. Different premotor neuron network models have been proposed. For instance, feedforward neural networks with a hyperbolic tangent activation function or a radial basis activation function were used to shape or translate oscillator signals into complex locomotion patterns [49,188] for climbing over or avoiding an obstacle [189,190]. To obtain motor memory for robust locomotion patterns, reservoir-based recurrent neural networks were applied [191]. While the typical technique uses various types of premotor networks and a learning method, recently Chuthong et al. [185] proposed an alternative approach that exploits the entrainment-like dynamics for CPG shape adaptation. This technique, called dynamical state forcing CPG (DSF-CPG), behaves as a reactive CPG that can temporarily adapt the geometry/shape of the CPG signals without using any learning or premotor networks. The DSF-CPG approach can promote robot compliance, by changing the target dynamics according to external perturbations.

For the phase adaptation, typically ground reaction force (GRF) feedback is employed to reset or continuously modulate the phase relationships between the oscillators. Two standard mechanisms for phase adaptation, resulting in adaptive interlimb coordination for self-organized robot locomotion, are phase resetting (PR) [181] and continuous phase modulation (PM) [159,182] (Figure 6a, upper inset). PR uses discrete GRFs to intermittently reset CPG phases while PM uses continuous GRFs to modulate CPG phases. A recent comparative study of the two mechanisms can be seen at [192]. Based on the experimental setup of the comparative study, PM shows slower but more stable phase convergence while PR shows faster but less stable phase convergence. PM performs better than PR when the robot is subjected to symmetrical GRF distributions while PR performs better than PM when GRF distributions are asymmetrical.

In addition to the CPG-based control described above, reflex-based control (pure reflexes), relying on sensory feedback, has also been widely used for robot locomotion generation with adaptability to different situations [29,59,118,193,194,195]. The most well-known reflex-based control is the Walknet control inspired by stick insect locomotion (see above and Figure 6a, middle inset). Walknet control is realized by using an artificial neural network (leg controller) for intralimb coordination and a coordination rule-based state machine for interlimb coordination. It has been implemented as a decentralized control architecture for hexapod robot locomotion [59,118] where six independent leg controllers are employed, one for each leg. Each leg controller requires position and velocity feedback of the joint angles of the leg as well as GRF and/or tactile contact feedback of the leg on the ground and consists of subnetworks for swing and stance leg movements. The subnetworks consist of a Stance-net, a Swing-net, and Target-nets. The Stance-net controls the movement during a stance phase. The Swing-net controls the movement during a swing phase. The Target-nets indicate the end position of the swing movement (AEP) during forward and backward walking. An extension of the original Walknet control has also been developed by adding motivation units for autonomously selecting between different locomotion behaviors [196]. Walknet control can generate emergent gaits for forward or backward hexapod walking over uneven surfaces. It can also allow for curve negotiation and leg amputations, and follow motion trajectories without explicit pre-calculation. The most recent version of Walknet (neuroWalknet controller) can be seen at [143]. Walknet control has also been modified (called Rollnet) for a ball rolling gait of a dung beetle-like robot [195]. The gait combines backward walking of the front legs and ball manipulation by the middle and hind legs.

While Walknet control shows impressive performance for versatile and adaptive robot behavior generation (locomotion and object manipulation), it may lead to unstable robot behavior or failure in cases of sensory failure. Therefore, a combination of CPG- and reflex-based control has been actively investigated and various types of this combination have been developed [142,197,198,199,200,201,202,203]. For instance, CPG-based control with a sensory event mistiming detection method and reflexes was proposed [201]. The mistiming detection method consists of a CPG for estimating the sensory phase, a radial basis function (RBF) neuron for estimating the sensory event, and a leaky-integrate-and-fire neuron for detecting the sensory mistiming and activating reflexes to avoid an obstacle and search for a foothold. This control approach enables a hexapod robot to walk effectively over highly unstructured terrain with cracks and a wet slippery surface. Adaptive neural locomotion control, consisting of a CPG network with neuromodulation and local leg control mechanisms based on sensory feedback (only foot contact) and adaptive neural forward models with efference copies (copies of CPG signals), was developed [142]. This control approach enables a hexapod robot to perform a multitude of different walking patterns, including insect-like leg movements and gaits (Figure 8), with energy-efficient locomotion. It can deal with changes in terrain, a loss of ground contact during the stance phase, stepping on or hitting an obstacle during the swing phase, and leg amputations. This robot can still perform basic locomotion even without the sensory feedback. Recently, an artificial hormone mechanism (AHM) was applied to the adaptive neural locomotion control [202,203]. AHM, which replicates the endocrine system, can continuously online adapt neural locomotion control parameters (lifelong continuous adaptation) for walking on different complex terrains (e.g., mesa terrain, ramp-up and -down terrains, rough terrain, terraced terrain, compliant terrain with different softness levels, and loose terrain).

#### 3.2.2. Engineering-Based Control

A conventional way of achieving intralimb coordination is to use inverse kinematics (IK) requiring a robot kinematic model [66,164] (Figure 6b). In this approach, the trajectory of the end of a foot (consisting of stance and swing phases) is designed and the IK translates the trajectory into the robot joint angle. For the trajectory design, one simple way is to use a straight or almost straight profile for the stance phase and an arch profile and swing phase [159]. An alternative way is to record an animal leg trajectory during locomotion and use it as the desired robot leg trajectory [66,126,204]. For instance, Cruse and Bartling [204] recorded and analyzed the swing trajectory in different stick insect walking situations and applied IK to calculate joint angles for robots. Ignasov et al. [66] introduced a complete framework for generating complex insect-like leg movements. The methodology consists of (1) tracking insect foot tip positions during walking; (2) simulating a bio-inspired robot with IK implementation; (3) transferring to a real robot; (4) validating robot foot tip positions. The insect foot tip positions and trajectories are recorded and analyzed frame by frame with a video tracking tool (e.g., Tracker). These trajectories are then used as the desired target positions which are converted into robot joint angles through IK for intralimb coordination. Ignasov et al. applied this framework to generate complex dung beetle-like leg movements during locomotion, dung ball manipulation, and dung ball transportation for a dung beetle robot prototype.

For interlimb coordination, this can be done by pre-defining the phase relationship between the legs for specific gaits. The basic setup will lead to certain locomotion patterns. To achieve dynamic locomotion or adaptation to uneven terrains, additional posture control mechanisms and environmental models are applied. For instance, the environmental model is employed to identify the roughness property of the terrain and obstacle information in such a way that the control system is able to plan and to adapt the foothold position, robot posture, and leg trajectory. This method requires exteroceptive feedback to perceive environmental information, such as that provided by a laser scanner sensor [205,206] and a depth camera sensor [164,207,208]. Recently, advanced engineering control methods like model predictive control and robot model abstraction-based control with planning, have been applied to achieve locomotion adaptation. For instance, Hu et al. [209] introduced a constrained model predictive controller for stabilization of non-periodic trajectories for walking, by a hexapod robot, over irregular terrain. Buchanan et al. [207] proposed a deformable bounding box abstraction of the hexapod robot model with mapping and planning strategies for hexapod locomotion control and body posture adaptation to navigate in confined spaces.

Insect-like muscle (compliant) properties can be achieved by a variable admittance/ impedance control [210,211]. However, it is a challenge to simulate coordinated compliant joint motions in adaptive insect-like locomotion due to high system redundancy [212]. These motions are achieved by integrating neural networks and muscle-like mechanisms [114,213]. To address this challenge, neuromechanical models have been developed to achieve coordinated compliant joint control on insect-like robots [214]. Szczecinski et al. proposed a novel way to tune a robot leg servomotor to exhibit insect muscle-like dynamics of equilibrium, perturbed responses, and active motions [215]. Ribak highlighted muscle compliance functions in insect and robot jumping [216]. He also provided potential insect-inspired solutions to solve small robot jump control and stability challenges. For instance, they bypass the power constraint of muscles by converting muscle work to elastic potential energy. Elastic elements act as springs storing the muscle work performed before jumping. Huerta et al. proposed an online muscle-like compliance adaptation control for robust 18-DOF insect robot walking [113].

#### 3.2.3. Machine Learning-Based Control

In addition to the bio-inspired and engineering-based control approaches mentioned above, an alternative way to automatically generate robot gaits for walking on irregular terrains and dealing with complex situations (like leg damage) is to apply machine learning (Figure 6c). To date, different machine learning techniques have been actively explored for locomotion control. The techniques include (deep) reinforcement learning (RL) [165,217], imitation learning [218], intelligent trial and error [219], and evolutionary computation [166,220,221].

For instance, Hafner et al. [165] developed a reinforcement learning framework that can learn locomotion behavior for different types of legged robots including hexapods. This framework relies on a data-efficient, off-policy multi-task RL method and simple reward functions. The learning time is approximately 5 h to obtain basic locomotion skills by walking in different directions. Ting et al. [218] proposed an imitation learning method that can train a “student” hexapod to imitate the walking behavior of an “expert” hexapod by watching its leg movements. Cully et al. [219] proposed an intelligent trial-and-error algorithm that allows a hexapod robot to automatically learn a behavior map consisting of over 13,000 high-performing behaviors or gaits. The process required 20 million iterations (roughly 2 weeks) on one multi-core computer to obtain the map. The robot can later use the map to search for an appropriate locomotion behavior to compensate for unexpected damage (like damaged, broken, and missing legs). Parker [220] used a cyclic genetic algorithm (CGA) to evolve control programs for generating different gaits of fully capable and damaged hexapod robots. Cully and Mouret [221] introduced the transferability-based behavioral repertoire evolution algorithm (TBR-Evolution) that can find several hundreds of simple locomotion controllers (one controller is for one possible walking direction). The algorithm relies on novelty search with local competition (searching for high performing and diverse solutions), and the transferability approach (integrating simulation and real tests to develop the evolutionary controller of a real robot). This approach enables a hexapod robot to learn to walk in every direction with a single run of the evolutionary algorithm within 3000 iterations (2.5 h).

Several studies have used artificial neural networks with data-driven methods for learning locomotion [222]. For instance, Azayev et al. [222] proposed a scalable two-level architecture for hexapod locomotion on complex terrain by using joint angle and binary foot contact feedback and deep reinforcement learning. The control approach can enable a simulated hexapod robot to navigate over ground, stairs, or through narrow pipes, etc. For fast complex robot locomotion generation, some studies have demonstrated the use of a combination of multiple control approaches where bio-inspired control and/or engineering-based control is used to encode basic rhythmic pattern generation while machine learning is applied for optimizing locomotion control parameters to achieve complex patterns (Figure 6). For instance, Milicka et al. [223] combined a chaotic CPG with IK (Figure 6a,b). In this approach, the CPG generates desired trajectories (foot-tip positions) while IK translates the foot-tip positions into the joint angles to directly control the actuators of a hexapod robot. As a result, the robot can perform a variety of movements, such as spot turning and walking with various gaits (e.g., tripod, ripple (a type of tetrapod), low gear (alternating between swinging two legs and one), and wave gaits). Chen et al. [224] used the same strategy combing a CPG with IK and additionally introduced force feedback to deal with irregularity in rough terrain (Figure 6a,b). While a combination of bio-inspired and engineering-based control can encode rhythmic patterns for a variety of hexapod gait generation and terrain adaptation through sensory feedback, one challenge of using such an approach is control parameter tuning or optimization. To address this issue, Fu et al. [225] proposed a combination of deep reinforcement learning with IK. The reinforcement learning method can automatically find motion planning policies for hexapod robots moving on uneven piles of plum-blossom (Figure 6b,c). Schilling et al. [217] introduced a biologically-inspired decentralized control architecture with deep reinforcement learning for adaptive locomotion in a hexapod robot. The architecture consists of six neural control modules, each of which controls one leg (Figure 6a,c). Thor et al. [188] presented a generic locomotion control approach, which combines bio-inspired CPG and RBF-based premotor neuron networks into a modular CPG-RBF neural control network (Figure 6a,c). This network uses a neural basis to produce complex rhythmic trajectories for the joints of walking robots. These trajectories are optimized using a probability-based black-box optimization (BBO) method. The framework was applied to teach and control three different simulated legged robots with varying morphologies including broken joints. Ouyang et al. [226] proposed an adaptive locomotion control approach which combines bio-inspired, machine learning, and engineering-based methods (Figure 6a–c). Specifically, they used a 3D two-layer CPG network where the first CPG layer for interlimb coordination generates basic robot locomotion patterns based on kinematics analysis. The second CPG layer is for intralimb coordination and controls the robot’s limb movements to deal with environmental changes. An actor-critic reinforcement learning method with deep neural networks (DNNs) (known as Deep deterministic policy gradient (DDPG) [227]) was employed to optimize the CPG control parameters of the second layer for fast and stable hexapod locomotion with adaptability to different terrains (flat, sand paper, soft sand, and 10 degree up-slope).

## 4. From Locomotion to Cognition

In the previous sections of the manuscript, the relevance of locomotion generation and control aspects, mainly related to biomechanics (see Section 2) and neuron motor activity (see Section 3), has been reviewed. Locomotion has the objective of making the body flexible enough to face the environmental challenge of reaching specific destinations. In other words, locomotion involves physical interaction with the environment realized through the physical motion of the body towards satisfying specific needs. On the other hand, motion, in general, is not separated from mental activity. It could be stated that any movement has its origin from a mental state which, constrained by the environment, it’s own possibilities, and by the internal motivational drives of the subject, designs, plans and provides the suitable commands for the execution of the corresponding action. Cognition (Figure 1a) involves non-physical (mental) interaction with the environment; once a certain mission has been defined, cognition refers to the “capability of planning ahead” i.e, of defining, from a high level, “what to do” and “how to do” in order to fulfill the mission [228]. This represents the main driver for the locomotion system, except for those actions purely driven by reflexes as also depicted in Figure 1. So, from this perspective, cognition is a brain function involved in planning body actions to reach complex goals. Centuries of research on mammals has tried to discover the details of how motion plans are conceived in the brain cortex to generate and plan limbs actions. Although this research has led to a number of important breakthroughs, especially in the neurorehabilitation field, we remain far from mastering the details of such a complex structure as the mammal brain cortex.

So, from a true engineering perspective, it will be more rewarding, at least in the near future, to work on those brain structures which are less difficult to study in detail, but which, at the same time, are able to generate complex behaviors that could boost the capabilities of current robotic structures. Within the animal kingdom, insects are the class of living beings most present in nature, thanks to their adaptation capability to extreme environmental conditions and their brains whose structure efficiently manages to exploit all of the body’s capabilities.

Only recently have insects been recognized to possess a “cognitive brain” structure [229]. In the interesting yet provocative work from Chittka [230], written in the last decade, it is argued that even simple brains possess all the ingredients necessary for most of the behaviors usually ascribed to larger and more evolved brains [231]. The idea started to arise that insects are not only reflex-based automata. In fact, even looking at simpler, non-social insects, they show, individually, such surprisingly complex behaviors as numerosity [232], attention and categorization-like processes [233], delayed matching to sample task (considered as the capability to learn the concept of sameness) [234], water maze solution [235], and other capabilities that can be defined as “proto-cognitive”. These are commonly considered as clear traits of high-level deliberative behaviors. The neural structures responsible for these capabilities, in some cases, can be investigated and reproduced in detail because they are so much simpler than those of mammals, and with a much lighter organization. To provide some examples, observational learning is among the functions likely to be important in future robotic research: insects, like wood crickets, are able to learn from the life-saving actions produced by other conspecifics when facing predators [236]. This clear example demonstrates how precious the result of this form of learning is, without it, survival would be impossible. It is well known that insects possess high level learning, called operant conditioning, which makes them able to learn on the basis of an expected outcome, under the guidance of a reward or punishment. This also means that insects can develop an expectation about the outcome of their actions, which can lead, in turn, to the planning of a specific behavior, or even a sequence of behaviors, even in the absence of the stimuli associated with the outcome. This form of expectation suggests lower and higher cognitive processes working together, and this has implications, for instance, in navigation and waggle dance communication in honeybees.

To include further key examples, bees learn a sequence of landmarks as cues for turns toward the feeder [237]. Thus, they develop expectations along a route associating specific landmarks in specific places within the sequence to the food source location. Route selection was also found to be time-dependent (e.g., morning or afternoon) [238]. Bees also perform novel shortcuts between different locations within a previously explored environment [239,240], but they also fly along shortcuts between a learned location and a location communicated by the waggle dance of a hive mate, without references to landmarks [241,242]. Observational learning even at a symbolic level is exemplified by dance communication in bees. Another important characteristic in insects is their capability to generalize, i.e., to flexibly respond when the animal is confronted with distorted versions of the learned associations, due to noise and developmental changes. Essentially this involves assessing the similarity between the presently perceived input and the previous experience. This generalization property was found in different sensory modalities such as olfaction [243], vision [244], and gustatory sense. Insects are also able to categorize, i.e., to group different objects or events based on common features [245]. Another important trait of insect cognitive capabilities, as introduced above, is “concept learning”, which relies on relations between objects, such as “same as”, “different from” [234], “above/below” and “to the left/right of” [246]. Insects are able to learn and also transfer the results of such learning also to unknown objects which may sometimes be completely different from those used for learning the relationships. More recently, bees were shown to process two concepts simultaneously and combine them in a rule for subsequent choices. This implies an even higher level of cognitive sophistication than dealing with one concept at a time.

The characteristics of functional/structural specialization of either side of the brain/ body (i.e., lateralization) involve a fascinating research activity carried out in different insect species, such as locusts [247] and beetles [248]. In Romano et al. [248], a biomimetic animal replica of a larger grain borer was developed to modulate the behavioral responses in insect. The results demonstrated that beetle pushing behavior is a complex communication strategy where the information is mediated through the performed actions in terms of number of acts and display duration. The desert locust is another well studied species as demonstrated in [249] where the generation of adaptive responses, such as evasive maneuvers was investigated in the presence of unpredictable events during a goal-oriented behavior, followed by a reorientation and route correction. This behavior is strongly related to selective attention in honeybees and flies where sensorial stimuli are opportunely gated by the nervous system depending on their relevance [250]. Social learning is another relevant characteristic ascribed to the cognitive repertoire of insect species such as ants, bees and wasps. In [251], the capability of social wasps to learn the facial features of all colony members was demonstrated with the aim of identifying the colony’s hierarchy as it relates to fighting skills. Some forms of social learning, commonly considered as high-level cognitive processes, are also reported in solitary insects species, such as crickets [252] or fruit flies [253] and recently have been shown also using robotic trainers [254]. This further outlines the added value in inspecting such forms of complex interaction capabilities in simple brains, from the engineering perspective aiming at building an insect brain model. Unfortunately, although these astonishing capabilities have been found in such tiny brains as those of insects, so far, the vast majority of them has not yet been specifically linked to the exact brain area which gives rise to such behaviors. For this reason, in the following section, we will refer primarily to those behaviors explicitly referring to the identified brain areas. This is mostly found in the case of *Drosophila melanogaster*.

### 4.1. The Fly Brain and Cognition

Among insects, *Drosophila melanogaster*, also known as the fruit fly, is considered as a primary model organism, widely studied to understand the details of the emergence of specific behavioral phenomena and draw rules that can be extended to higher organisms. The fruit fly is a perfect candidate thanks to the low number of neurons present in the central brain ($10^{5}$) as compared to other well-studied animals like the brown rat ($2\times 10^{8}$) and human ($8.6\times 10^{10}$). Another important aspect is the possibility of creating mutants using genetic tools based on the GAL4-UAS technique [255]: the acquired knowledge of the *Drosophila* genome allows the manipulation of the nervous system of a fly to study the structural and functional relationships between its neural structures and the different behavioral capabilities.

To focus our work on the architecture where a clear picture of neural circuits, behavior functionalities and experiments (e.g., either in simulation or using actual robotic systems) was available, in the following section we will mainly deal with the *Drosophila* brain and concentrate our attention on its behaviors related to walking.

Intact adult *Drosophila* are naturally able to fly, and do not demonstrate relevant walking capabilities; but if their wings are clipped, a lot of experimental results have testified to their capability of demonstrating adaptive locomotion skills. These have to be learned from scratch, demonstrating the impressive flexibility of such a tiny brain to adapt to a novel ecological niche.

In Section 4.2, we will introduce the insect brain architecture describing the neuroanatomical structure of two relevant neuropiles in the central brain. In Section 4.3, we will introduce the state-of-the-art on biological experiments, functional models, and robotic applications in relation to the neural structures involved. In Section 5.3, we will present some future directions in insect-inspired robotic cognition.

### 4.2. The Insect Brain Structure

The *Drosophila* central brain includes two important neuropiles: the mushroom bodies (MBs) mainly devoted to the adaptive termination of behaviors and responsible for olfactory learning and multimodal integration, and the central complex (CX) responsible for the initiation of behaviors and mainly involved in processing visual stimuli.

MBs are a paired structure present with similar forms in all insect species.

The intrinsic neurons of the MBs are called Kenyon cells (there are about 2000 KCs per hemisphere in the fruit fly) which receive predominantly olfactory input in the calyx region. From there they project through the peduncle into different lobes (i.e., α−/β-lobes, $\alpha^{\prime }-/\beta^{\prime }$-lobes and γ-lobe). The first identified function of the MBs corresponded to olfactory learning and memory. Olfactory receptors mainly reside within the antennae where olfactory receptor neurons (ORN) transfer the sensory information to the MBs to build associative memories for odors.

Odor perception and processing are fundamental because flies identify food sources and select sex partners through this sensory system. Although input from other sensory modalities is anatomically not evident in *Drosophila* MBs, there are several experiments showing the role of this center in tasks related to vision [256]. The bee brain, in turn, shows the presence of gustatory, and mechanosensory inputs in the MBs [257]. The MBs can therefore be ascribed as a multisensory integration system where mixed-modality signals are collected together with negative and positive reinforcement information conveyed by dopaminergic and octopaminergic neurons. The use of genetic tools has enabled the identification of other relevant functionalities associated with the MBs: it has been shown that mushroom-body-less flies have problems switching to a new type of behavior in the presence of changes in the environment; MBs are also devoted to sleep control and they form the basis of motor learning [258].

The CX neuropile resides in the middle of the MBs-paired structure and, in *Drosophila*, can be divided into the following substructures: the protocerebral bridge (PB), the fan-shaped body (FB), and the ellipsoid body (EB) (Figure 9) [259].

The PB is responsible for step-size control, selecting the proper speed, and direction of motion as demonstrated in walking and gap-climbing scenarios. Experiments performed using a flight simulator revealed that the FB in *Drosophila* is responsible for visual memory functions, extracting visual features that can subsequently be associated with specific behaviors. We can summarize the role of PB and FB as the centers responsible for answering the questions: “where?” and “what?”. Finally, the EB plays a relevant role in orientation memory and place learning. Its ring structure is able to identify the angular position of the object of interest and participate in the path integration process.

Even if, in *Drosophila*, there is no explicit involvement of the MBs in such tasks as visual learning, in higher complex structures, like ants and bees, MBs are definitely multimodal structures, hosting both olfactory and preeminent visual stimuli [260].

A block-size model of the different neural centers and their interaction with the sensing and actuation areas is reported in Figure 9. The sensory modalities taken into consideration are vision, olfaction and touch. The role of the MBs and CX is indicated in terms of processing the different sensory stimuli and enlightening the interactions among the relevant neuropiles we have considered. The motor command circuits are placed within the thoracic ganglia where local reflexes are also implemented. Rewarding and punishing signals mediated by octopaminergic and dopaminergic neurons are used as a driver for the learning processes.

### 4.3. Insect Brain Functional Models, Implementations and Robotic Experiments

From the analysis briefly outlined above, we can deduce that each of the two main centers, MBs and CX, have individual functional roles, but there are complex tasks that require the concurrent enrollment of both. In this subsection, these main functionalities will be reviewed and shown via both computational models and experimental robotic architectures. A summary of the state-of-the-art in the field is depicted in Table 3.

#### 4.3.1. MB Models

Insects, with particular reference to bees and ants, are famous for homing behaviors: they are able to memorize a visual route that allows them to come back to the nest, even after navigation covering several hundred meters from home. In the study [260], the authors propose that this ability can arise from the interconnections among KCs in MBs, which could be in charge of encoding a spatiotemporal memory of visual motion while moving along a spatial route. Here a fairly reproducible computational model, demonstrating such behavior, is presented in detail. The model internally builds causal correlations among subsequent visual events, thus creating a short sequence learning structure. The model represents a simplified ant MB architecture employing a modified leaky integrate-and-fire neuron model and the Spike-timing-dependent plasticity (STDP) learning rule [284]. The neural response shown by the model matches the observed behavior in ant experiments. The model was finally demonstrated in action, performing indoor navigation control in the dual drive wheeled robot named TurtleBot Burger3 equipped with an event-based camera. The model presented above has a lot of similarities to the one developed some years before in [268,269], exploiting the deeper details known in the fly brain. In fact, the discovery of axo-axonal connections within the MB peduncle allowed the modeling of the phenomenon of sequence learning. It is a complex spatial temporal correlation task, where waves of consecutive stimuli, consolidated via reinforcement signals, led to the formation of complex sequences and sub-sequences of abstract stimuli. Different types of neuron structures and learning schemes were adopted in the overall architecture: Izhikevich’s spiking neurons as well as Morris–Lecar models, or STDP and Hebbian-like learning for the different layers. The model was successfully tested on a differential drive robot trained in a reinforcement environment. One added value of the developed structure is the capability of internal simulation of the correlations learned during the experiments, which can explain the phenomenon of overnight memory consolidation. In fact, the network is able to reproduce the visual stimuli, simply as an effect of internal noise. These stimuli can create the internal dynamics whose effect is to reinforce the previously learned sequences. Part of this structure was subsequently found to act as a suitable neuromorphic model for classification and decision control [270]. Starting from the experimental evidence as well as from the known architecture of the fly brain, some additional behaviors were hypothesized to be hosted, which were already found in more complex insects like bees or ants. In particular, the role of extrinsic neurons connecting the lobe systems to the antennal lobe, paired with Hebbian-like learning within the antennal lobe system, was computationally found to be responsible for the elicitation of the attentional loop [264]. The model was tested in an experimental arena involving a robot that was able to maintain a targeting behavior while filtering out other distractors. As it developed, this model became able to learn what would be considered as an expectation, i.e., the capability to predict the appearance of a given (expected) input stimulus, if preceded by another (triggering) one [266].

Another relevant phenomenon that was found in flies is related to decision making when facing visual dilemmas: once the fly learns that a given association color-shape is rewarding, when faced with a dilemma caused by altering the color-shape arrangement of rewarding and punishing visual stimuli, the intact insect has a clear preference for the color. Once the color saturation goes below a certain threshold, flies suddenly decide to choose the shape learned as rewarding. This is a clear trait in decision making, which is experimentally ascribed to the MBs. A possible computational model was introduced in [271].

The MB structure, as in the case of attention [264], was also found able to host high-level behavior learning such us the so-called “delayed-match-to-sample task”, considered as similar to the acquisition of the concept of sameness and difference [267].

#### 4.3.2. CX Models

The CX of an insect participates in processing the temporal and spatial components of sensory cues, and utilizes those cues in creating an internal representation of orientation and context, while also directing motor control. In addition to the fruit fly, the role of the CX in controlling goal-directed navigation has been assessed in other insect species, such us cockroaches [275], finding a lot of similarities with navigation in higher animals (e.g., rats). In the study [275], an overview of current knowledge on the CX’s role in insect navigation is performed. A deep study revealed that the apparent random path ending in the finding of a dark shelter was indeed a targeting behavior toward the shelter. To prove this, in the study [276], a computational model (called RAMBLER) was built after a series of experimental campaigns. It efficiently models the insect’s motions, combining the classic wall following behavior with the visual verification of the dark shelter and the probability of adding other motions, such as turning back, whenever the shelter appeared behind the insect. Although the model was obtained exclusively by using insect experiments, several factors imply a direct responsibility of the CX in such tasks. For example, CX lesioned roaches often make wrong turns, and also speed changes while walking are preceded by an increased firing rate in CX neurons.

One of the main CX roles is to host visual and mechanosensory stimuli to learn such characteristics as body size, i.e., the awareness of the peripersonal space. Genetically identical flies, if fed abundantly, can grow up to 20% larger than normally fed ones. If differently sized flies are about to engage in a fight, the smaller fly gives up without fighting. Body size knowledge was discovered to have been learned during larval stages, if flies are grown in well-lit environments. They learn the association step-stride versus distance walked, creating a conceptual function describing “how big they are”. This function was found to be resident in a sub part of the CX, known as the protocerebral bridge. A computational model of this behavior was designed and implemented in a simulated hexapod robot [35], and finally tested using a humanoid robotic platform [278]. This proves that computationally relevant models, drawn from insect structure, can also be adopted for controlling different robotic bodies, not necessarily related to an insect-like biomimetic structure.

To try to shed light on how the CX exploits sensory inputs to realize motor functions associated with spatial navigation, in [285], a neural model was developed, based on known connectivity within the CX with particular attention to the Ellipsoid Body (EB). The model was based on continuous-time differential equations called leaky integrators, used to simulate the mean-field activity of pools of neurons, and used the Monte Carlo method for parameter estimation. The model was successfully tested by means of a kinematic simulator. The same leaky integrator model was employed to generate robust ring attractor dynamics [274], i.e., neuron structures, which responded to the position of external stimuli and retained activity even in the absence of external stimuli. In that work the authors hypothesized that the emergence of ring attractor activity in the EB is due to reciprocal neuronal connections between EB and PB. Ring attractor networks were also modeled using spiking neuron structures. These were found to be able to show spatial memory formation exploiting the ring architecture of the EB [273].

#### 4.3.3. Models Involving MB-CX Interaction

Some behaviors require, at least at the functional level, the participation of both MBs and the CX. This is the case, for example, in navigation tasks that are efficiently performed by insects using visual sensory feedback. Insects typically show innate curiosity for landmarks, especially moving ones, towards which they are attracted. Such landmarks can be used to elicit, after learning, specific directional steering maneuvers, which can subsequently be understood to be part of a complex navigation task. While the neuronal details of the concurrent existence of these two aspects of visual learning are not yet known, a potential computational model was proposed in [280] based on the neuroanatomy of the wood ant CX, typically involved in landmark targeting behavior. Here, a reward signal, supposed to be provided either by the innate CX landmark attraction feature or by a long-term visual memory, built into the ant MBs [260], was used to form a local vector memory in the CX. The strategy was implemented using a simulated robot in a simple environment endowed with a single visual cue. The architecture matches the experimental results coming from unilateral MB lesions in ants, which restore the innate targeting behavior.

In the study [192], to address insect navigation in a cluttered environment, the authors speculated that specific strategies are needed to retrace familiar routes and return home. These would need both the CX and MBs working together: MBs check whether the current sensory stimulus is positive, while the CX steers the animal’s heading towards the rewarding stimulus. The model was found to reproduce behavioral data in realistic environments.

Another experiment, called detour, was performed to demonstrate the presence of short-term visual memory in the fruit fly. The experiment verified that the fruit fly can show short term redirection towards invisible targets after the introduction of a second landmark, called a distractor [272]. This fact reinforces the idea that very similar behaviors, typically found in higher animals, can be shown in simpler brains, which, from the engineering perspective, can be more efficiently modeled.

As stated above, the CX is involved in spatial learning from visual cues as demonstrated in *Drosophila* [235]. Of course, larger and better CX structures in bees were ascribed possible roles in the spatial learning of color cues. Moreover, in the study [283], using a combination of color learning with electric shock as punishment, an interplay between CX and MBs was found by pharmacologically silencing these two centers. In particular, the CX was found to be necessary for mediating the goal-directed behavioral response to learned stimuli, while the MBs carried out the actual cue association.

As discussed above, wing-clipped flies have to learn from scratch how to efficiently interact with the environment. Therefore, they must adapt their locomotion capabilities to more complex situations. A joint study with neurogeneticians, supported by experimental evidence, led to the conclusion that the fly, responding to a difficult climbing scenario, started by making some attempts, and the most rewarding ones are retained and adopted to build the right sequence of leg motions for fulfilling the climbing action. A corresponding model, involving the concurrent activation of MBs and CX, was also developed [282].

## 5. Lessons Learned from This Review

### 5.1. Future Directions in Biomechanics

Emerging biomimetic leg design will be increasingly based on available micro-CT scans of real insects in order to improve their level of complexity, fidelity, and bio-inspiration in the 2020s. So far, none have been built with a structure similar to an ant exoskeleton. This kind of structure gives ants the ability to load and transport up to 15 times their own weight.

Many aspects of force sensing in insects could be applied to hexapod robotics, and these will be particularly helpful in redesigning legs which incorporate force sensing. Moreover, adding some compliancy to the leg structure at strategic locations could provide force sensing, and significantly reduce the relative mass of legs in relation to the robot’s overall weight. In insects, force measurement is done at the level of the *campaniform sensilla* [286]. The *campaniform sensilla* are mechanoreceptors providing local control of legs; some current mechanical solutions mimic them by adding a slider and a spring per joint, but at the expense of compactness and increased mass.

### 5.2. Future Directions in Locomotion Control

The present review on locomotion control shows that significant efforts have been made to understand the principles of insect locomotion control and translate those principles to robot locomotion control using different technical control methods such as bio-inspired, engineering-based, machine learning-based, as well as these methods in varying combinations. However, current robot performance is still far from that of insects. This is due to generated complex behavior that results from emerging processes derived from biomechanics (see Section 2), locomotion control (covered in Section 3), and high-level cognitive control (see Section 4) as well as their dynamical interactions (Figure 1). Thus, focusing on particular aspects can only achieve partial solutions. From this point of view, future research directions might endeavor to integrate all these components and exploit their dynamical interactions. Such an integration can be viewed as a complete embodied closed-loop system [287]. This will also shed light on a key research question: *how do low-level motor/locomotion control and high-level cognitive control mechanisms (brain), a body (sensory-motor system), and an environment dynamically interact in real time to generate motion intelligence?*

While insects can intelligently exploit all their body parts (i.e., legs, thorax, and abdomen) for stable and versatile locomotion and object manipulation, most insect-like robots use only their legs for locomotion and object manipulation. During walking and climbing, a cockroach tries to keep its abdomen close to the ground (i.e., low center of mass) [288]. Ants can also exhibit a kind of “clutch” behavior with their legs and bodies, to try to avoid being blown away by a gust of wind [289]. During dung ball rolling, a dung beetle uses a part of its abdomen as another contact point to maintain stability [144]. From this point of view, such strategies, which can be considered as whole body locomotion and object manipulation, should also be realized for performance enhancement in insect-like robots [142,290]. As legged robots have been actively integrated into people’s daily production activities and various service tasks (such as exploration, object transportation, inspection and maintenance, search and rescue, and construction), fast lifelong continuous adaptation for resilient and robust robot locomotion to deal with environments that change unexpectedly, and physical damage over the course of a mission, is becoming increasingly important.

### 5.3. Future Directions in Insect-Inspired Robotic Cognition

From the review just presented, it clearly emerges that, over the last decade a huge effort in the design of powerful targeted experimental setups has been able to describe interesting behaviors in insects. These ranged from the simple associative learning skills to surprisingly relevant behaviors that can definitely be considered proto-cognitive. Despite the relevance of these discoveries, engineers are still concentrating their efforts on the design and implementation of single experiment tailored models, both at the computational level and, sometimes at the robotic level. Although these steps are needed to address the necessity and sufficiency of specific neural assemblies’ roles in the generation of corresponding behaviors, both at the insect experimental level and on the robotic side, the modeling strategy is stuck at a block sized level. As also reported in [291], the insect brain is a prime site for the detailed study of the link between neuroanatomical geometries, the corresponding function and the final behavioral outcomes, all the way from a wiring diagram to the level of insect intelligence. One of the main open research questions regards the possibility of studying the insect brain no longer as a block sized architecture, but viewing it from a holistic perspective, i.e., looking at the “whole”, instead of considering simply a part of its composition. The pursuit of such a research line should result in greater benefits across a variety of fields such as neuroscience, bio-medicine, artificial intelligence, and robotics. Nowadays, a whole insect brain model is only partially achievable due to the limited knowledge both at the level of experiments on single behaviors and on the study of the brain as a whole. Indeed experimental data covers only a small part of the huge dynamical neural activity responsible for a plethora of concurrent sensory motor activities. The authors could justifiably claim that the constraints just cited, and the still inadequate computational resources constitute the current limit, but these are likely to be overcome by creating biologically realistic simulations of the entire insect nervous systems and this should be possible within the next 20 years [291]. Moreover, the authors believe that a possible “Rosetta Stone” would be to consider the role of emergence in such tiny brains. Huge dimensional nonlinear dynamics can emerge simply by wiring non trained neural assemblies together. This would be achieved through careful neural wiring developed by extracting the connecting rules from the insect connectome. In addition, trainable read-out maps could be used to extract the specific sub-dynamics from this neural lattice to create ad-hoc associations with different behavioral needs. This could be realized concurrently. This new line of research, drawn from psychology, is known as “neural reuse”: the same neural assembly can be exploited concurrently for different parallel tasks. Also, as outlined in [292], the smaller the brain, the larger the need for neural reuse. Insect brains are of a suitable size to try to investigate neural reuse in action, since in these small brains, the relatively low number of neurons and mainly short-distance connections are candidate elements for neural reuse, even if the circuits are composed of different brain areas. The authors believe that the neural reuse paradigm will be the best route to take to explain all the rich functionalities of insect brains, and could open the way to describing even more complex combinations of non trivial behaviors that characterize the efficiency of more sophisticated brain structures. In any case, even the replication of an insect brain on an insectoid body would provide an added value for attaining true autonomy in future advanced robotic structures.

Active research on insect cognitive capabilities could also be relevant to better understand the evolution of cognition in relation to the well-studied vertebrate field [293]. The next steps will include the definition of behavioral tests to measure cognition and the identification of the relevant factors that contribute to the evolution of cognition.

The authors are convinced that a complete understanding of the insect brain will give a considerable boost to both biological and robotic research, leading to the introduction of a new generation of adaptive, resilient, robust, and efficient bio-inspired machines.

## 6. Conclusions

Over the last ten years, the booming use of 3D printing has significantly boosted the development of brand new mechatronic designs for hexapod robots. So, what can we expect from the next 10 years in hexapod robot design?

New emerging printable materials combined with multi-material 3D printers will make it easy to design and to reproduce insect-based legs by using micro-CT scans of real insects in order to reach their level of complexity. This will, not only lead to a better understanding of how insects walk, but also to finding innovative leg structures incorporating smart force sensing by adding soft materials. Force sensing is a critical aspect in locomotion control for complex terrain prediction and adaptation, adding both compliance segments and actuators to the leg structure at strategic locations is the major issue for the next decade. In insects, force measurement is done at the level of the *campaniform sensilla* on their exoskeleton close to their joints, and not at the tip of their leg. Such an approach in force sensing could significantly improve the level of compactness of future hexapod robots. By means of such bio-inspired approaches to insect morphology coupled with 3D printing, leg structure based on exoskeleton design will be easy to prototype, to clone for academics, and to repair. Such a bio-inspired exoskeleton could permit a significant reduction in the legs’ weight, and therefore enhance the capacity of load transportation in hexapod robots.

Locomotion control is also an important element in achieving intelligent behaviors (e.g., adaptive locomotion, load/object transportation) in hexapod robots. It is a basic mechanism to realize interlimb and intralimb coordination as well as joint compliance. Biological findings reveal that there are at least three components (CPGs, sensory feedback, and forward models) supporting the control. Thus, these components should be taken into account when developing locomotion control for hexapod robots. CPGs basically act as oscillators that produce basic rhythmic patterns for joint movements and gait generation without sensory feedback (open-loop control). Sensory feedback, however, is important for the frequency, shape, and phase adaptations of CPGs which result in adaptive locomotion (closed-loop control). Furthermore, it also drives reflexes to allow for walking on difficult terrains. Forward models complement the control in terms of sensory prediction or state estimation for highly adaptability. While hexapod robotics has been mainly focused on locomotion control, future research should be moved towards controlling a variety of robot functions, approaching insect-like abilities such as object manipulation and transportation. This can be accomplished by considering how to make use of the entire robot body (i.e., legs and trunk) as insects do. Machine learning methods can also be used as part of the locomotion control method to create resilient and robust robot functions that can adapt to changing environments. Additionally, these methods, together with hexapod robots, can be used to decode insect movement intelligence, e.g., muscle multifunctionality in adaptive locomotion and object transportation.

As stated in the introduction, the details of the neural mechanisms in charge of each brain function, and how these neural mechanisms are coordinated for the implementation of complex tasks remain largely unknown in many cases. For this reason, a large number of operative hypotheses often need to be formulated to allow the design of efficient computational models. This task would definitely be facilitated by close collaboration between neurogenetics, computational neuroscience and robotics. The first, thanks to modern tools, is able to create mutants by targeting specific neural areas, addressing the behavioral role of specific neurons. The second would in turn design and build efficient computational, neurally grounded models, and the last would then implement these models in working robotic structures. Therefore, it is necessary to foster collaboration between these different scientific fields to extract from biology the necessary information to be translated into efficient deployable models and structures. At the other end, robotics offers the unique opportunity to see the computational insect brain models in action in mechatronic structures. This can generate new ideas for insect experiments. Moreover, experience suggests that the lack of suitable experimental setups has left a lot of insect behaviors undiscovered. In this sense too, a close collaboration with engineers would be invaluable.

Insect brains are much simpler than vertebrate brains, and so can be accurately modeled and reproduced. An insect brain could deliver, to an artificial body, capabilities of adaptation and autonomy, which would be called “insectoid”: a robotic structure endowed with an insect-inspired brain.

The cognitive architecture modeling the insect brain has been proven to be independent of the particular robotic structure, although it co-evolves with the robot body by adapting to the behavioral outcome compatible with that specific structure.

## Figures and Tables

**Figure 1 sensors-21-07609-f001:**
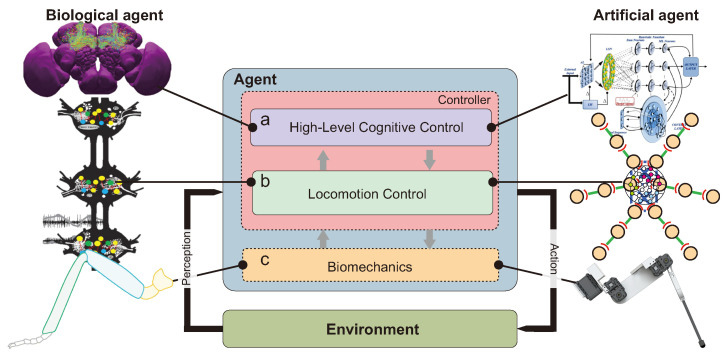
Overview of three main components underlying complex locomotion and cognition: biomechanics, locomotion control, and high-level cognitive control. Left (right) shows examples of the three components in insects (robots). (**a**) Left (right) is the insect central brain (high-level neural cognitive control model). (**b**) Left (right) is the insect thoracic ganglia (modified from [6]) (robot locomotion control model). (**c**) Left (right) is a biomechanical insect (robot) leg.

**Figure 2 sensors-21-07609-f002:**
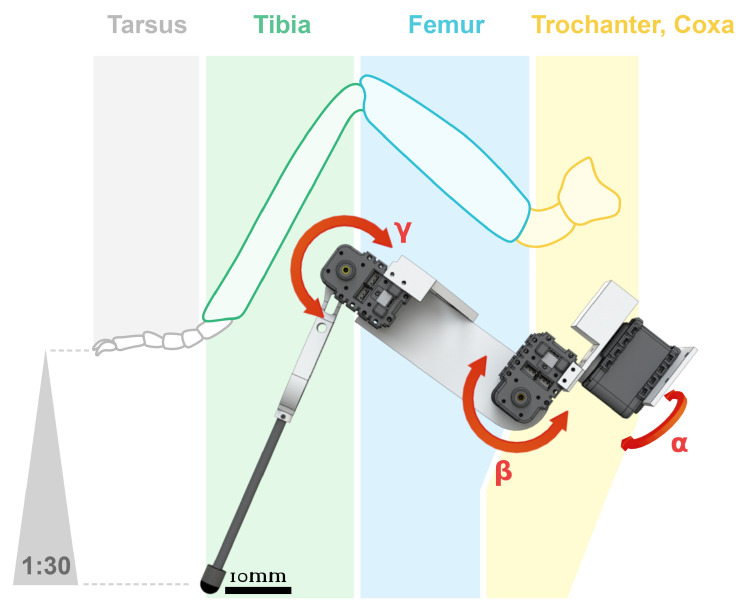
A standard hexapod robot 3-DOF leg, based the *Cataglyphis fortis ant* scale 1:30. Angle α corresponds to the thorax-coxa joint position, angle β corresponds to coxa-femur joint (up to now, robotic designs have often fused the trochanter-femur joint), and angle γ represents femur-tibia position. Illustration: ©Camille Dégardin & Ilya Brodoline (2021).

**Figure 3 sensors-21-07609-f003:**
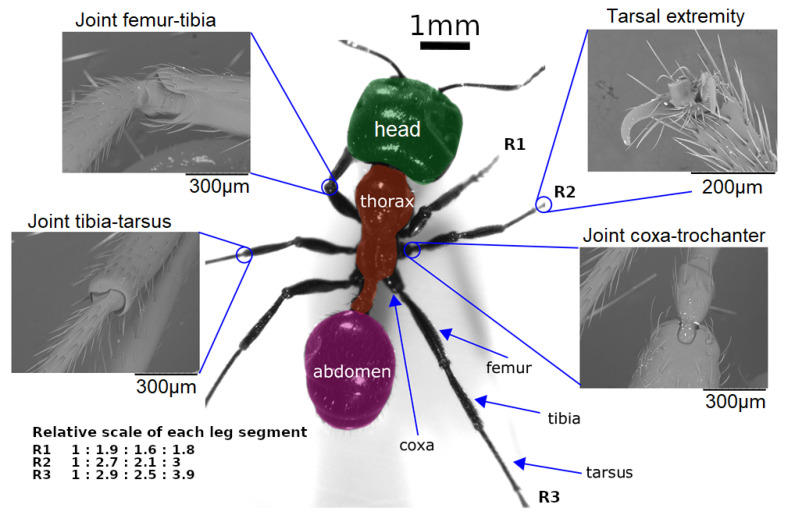
Scale, segments, and joints in ant *Messor barbarus*. Photographic credits: Hugo Merienne, Centre de Recherches sur la Cognition Animale (CRCA UMR 5169), Toulouse, France. The relative scale of each segment (coxa, femur, tibia, tarsus) w.r.t. the coxa leg of each leg (R1, R2, R3) comes from [73]. Adapted from [82] under CC-BY License, 2019.

**Figure 4 sensors-21-07609-f004:**
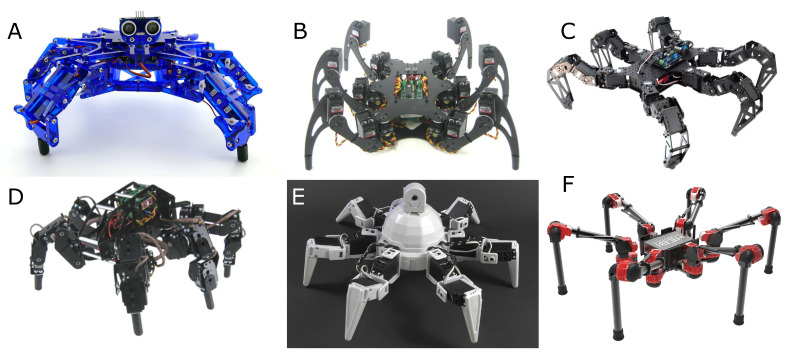
Open-source hexapod robots and educational robots available in kit form. (**A**) Hexy (Credits: ArcBotics). (**B**) Phoenix (Credits: Lynxmotion). (**C**) PhantomX (Credits: Interbotix Lab., Trossen Robotics). (**D**) T-Hex (Credits: Lynxmotion). (**E**) Six (Credits: EZ-Robot). (**F**) HEBI Robotics’ 18-DoF Daisy Hexapod, built using HEBI’s X-Series actuation hardware (Credits: ©HEBI Robotics 2021).

**Figure 5 sensors-21-07609-f005:**
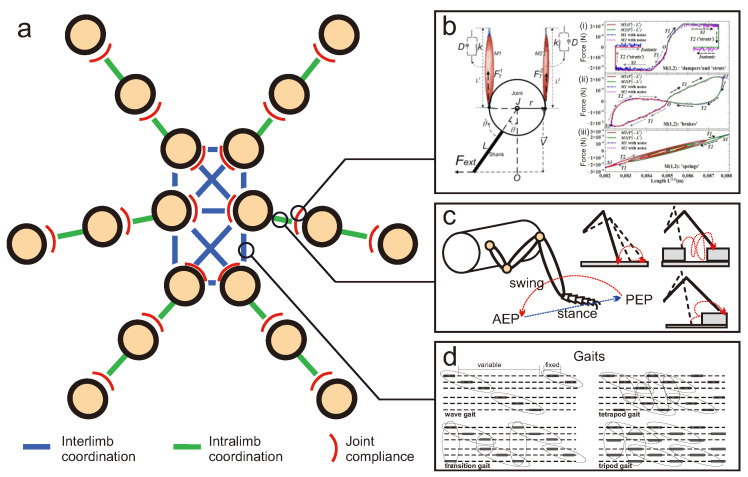
(**a**) Overview of locomotion control involving interlimb coordination (coordination between legs), intralimb coordination (coordination between joints in a leg), and joint compliance (a property of joint with variable stiffness). (**b**) Example of joint compliance generated by an adaptive bio-inspired muscle model (modified from [115]). Using this model, a leg joint can behave like dampers, struts, brakes, or springs (modified from [115]). (**c**) Example of intralimb coordination in the swing and stance phases of the leg and different reflexes for walking on uneven terrain (modified from [116,117,118]). (**d**) Example of interlimb coordination which results in different gaits (modified from [119]). The main observed gaits from slow to fast [120] include: wave gait (only one leg lifts at any given time (swing) while the remaining legs stay on the ground (stance), and the wave travels from back to front), transition gait (front and back legs of the opposite side lift together at a given time while the other legs stay on the ground), tetrapod gait (diagonal pairs of legs lift together at a given time while at least four legs stay on the ground), and tripod gait (front and back legs of one side and the middle leg of the opposite side lift together at a given time while the remaining three legs stay on the ground). While the articles specify specific gaits, it is important to note that certain insects (such as stick insects [121,122], cockroaches [123], and flies [124]) frequently change their gaits depending on their locomotion speeds and situations [125].

**Figure 6 sensors-21-07609-f006:**
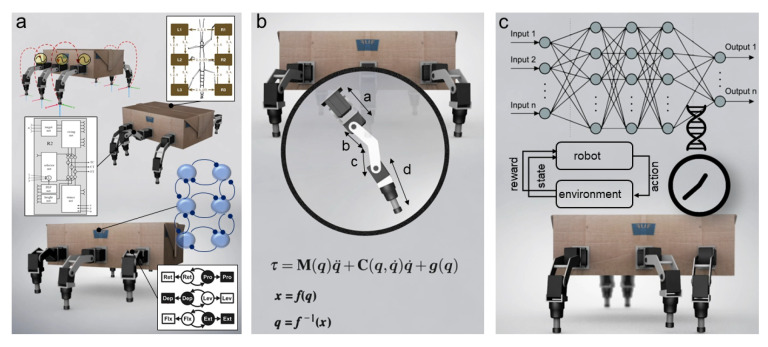
Different control approaches to robot locomotion. (**a**) Bio-inspired control. The upper-inset shows distributed decentralized CPGs with force feedback [158,159,160]. The middle inset shows the Walknet-control (modified from [143,161]). The lower inset shows the CPG-based control and a simplified model with six oscillators for interlimb coordination (modified from [162,163]). (**b**) Engineering-based control [66,164]. (**c**) Machine learning-based control [165,166].

**Figure 7 sensors-21-07609-f007:**
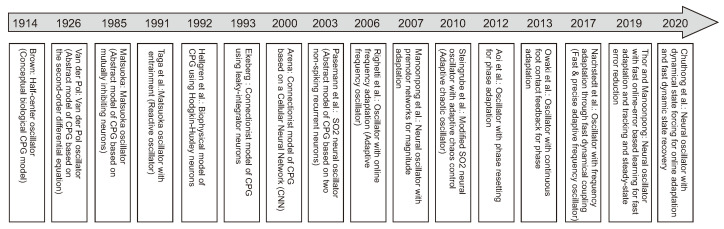
Timeline of the development of various CPG models from 1914 to 2020.The key CPG models include a half-center oscillator in 1914 [172], the Van der Pol oscillator in 1926 [177], the Matsuoka oscillator in 1985 [178], the Matsuoka oscillator with entrainment in 1991 [179], a biophysical oscillator based on Hodgkin–Huxley neurons in 1992 [173], a connectionist oscillator based on leaky-integrator neurons in 1993 [174], a connectionist oscillator based on a cellular neural network (CNN) in 2000 [175], an SO2 neural oscillator in 2003 [176], an adaptive frequency oscillator (AFO) in 2006 [180], a neural oscillator with magnitude adaptation in 2007 [49], an adaptive chaotic oscillator in 2010 [141], an oscillator with phase resetting [181], an oscillator with continuous phase modulation [182], an oscillator with frequency adaptation through fast dynamical coupling (AFDC) in 2017 [183], a neural oscillator with fast online-error based learning in 2019 [184], and a neural oscillator with dynamical state forcing (DSF) in 2020 [185].

**Figure 8 sensors-21-07609-f008:**
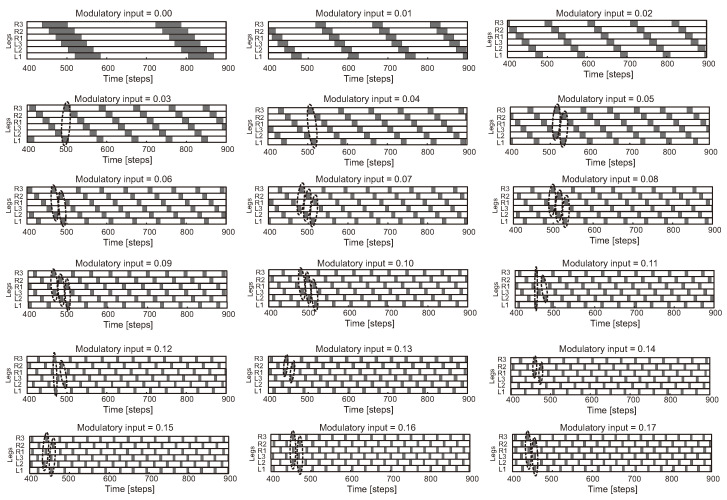
A variety of hexapod gaits with varying speeds generated by adaptive CPG-based neural locomotion control (modified from [142]). The frequency of the CPG outputs can be changed by modulating the synaptic connections of the CPG neurons with an extrinsic modulatory input MI. When the modulatory input MI is set to 0.0, each leg steps in a wave on each side with overlap. Stepping frequency increases as MI increases, and some legs step in pairs (see dashed enclosures). This results in insect-like gaits (Figure 5d) and various intermixed gaits. The caterpillar gait is characterized by the movement of two front, middle, or hind legs at the same time and the wave travels from back to front. Under this control approach, a transition gait as shown in Figure 5d is not found. For example, one can observe wave gaits with varying frequencies (MI = 0.01–0.04), tetrapod gaits with varying frequencies (MI = 0.05–0.06), caterpillar gaits with varying frequencies (MI = 0.07–0.10), and tripod gaits with varying frequencies (MI = 0.15–0.19). Legs are labeled as numbers 1–3 from front to back, and the left and right sides are L and R, respectively. It is worth noting that when MI is raised above 0.17, only two different gaits comparable to tripod gait (e.g., MI=0.17) and caterpillar gait (e.g., MI=0.10) appear.

**Figure 9 sensors-21-07609-f009:**
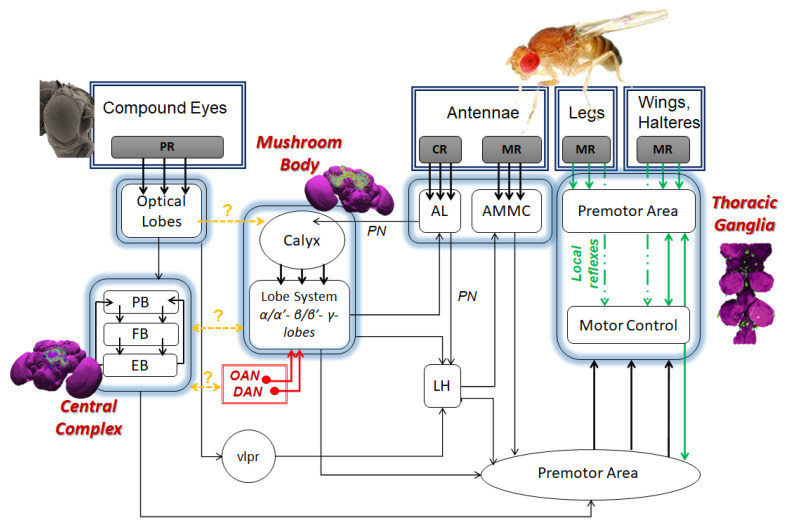
Block-size model of the relevant sensory modalities and neural processing centers in a *Drosophila* central brain. The insect’s compound eyes photo-receptors (PR) acquire visual stimuli that are initially processed in the optical lobes, and transferred to the CX, and then the neuropile responsible for visual orientation. Chemical-receptors (CR) located in the antennae are responsive to olfactory stimuli whose neural response is transferred through the antennal lobes (AL), and subsequently to the projection neurons (PN), and finally to the MBs. The olfactory and visual inputs are here integrated with the other sensory modalities although their connecting paths are still not evident. The lateral horn (LH) is an inhibitory center that is activated by the PN and the ventrolateral protocerebrum (vlpr) to affect the MBs activity. Dopaminergic (DAN) and octopaminergic neurons (OAN) provide reinforcement learning signals used by learning and memory systems. Tactile stimuli, acquired by the mechanoreceptors (MR) located in the antennae, legs and halters, are locally processed in the antennal mechanosensory and motor center (AMMC) and in the thoracic ganglia for the generation of local reflexes where the CX is also concerned.

**Table 1 sensors-21-07609-t001:** Overview of the state-of-the-art hexapod robots developed over the last 20 years in the range 1∼27 kg. The size, given in meters, corresponds to the largest dimension between width and height. DOF stands for Degrees Of Freedom for the entire robot (note that this can include extra actuation for head control and body control). The speed represents the maximum speed measured, in meters per second. The symbol “-” represents missing data.

Year	Ref.	Hexapod	Size [m]	Mass [kg]	DOF	Compliant	Speed [m/s]	Task
2021	[14]	HAntR	0.50	2.9	24	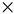	0.43	Locomotion
2019	[15,16]	MORF	0.60	4.2	18	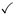	0.70	Locomotion
2019	[17]	Daisy	1.10	21	18	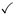	0.13	Locomotion
2019	[18,19]	Drosophibot	0.80	1	18	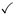	0.05	Locomotion
2019	[3,20]	AntBot	0.45	2.3	18	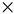	0.90	Navigation
2019	[21]	Corin	0.6	4.2	18	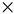	0.10	Locomotion
2018	[22]	AmphiHex-II	0.51	14	6	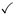	0.36	Locomotion
2018	[23]	CRABOT	0.70	2.5	24	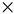	0.05	Locomotion
2017	[24,25]	PhantomX AX	0.50	2.6	18	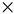	0.29	Locomotion
2017	[20,24]	Hexabot	0.36	0.68	18	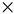	0.35	Navigation
2016	[26]	Weaver	0.35	7	30	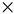	0.16	Locomotion
2016	[27]	MX Phoenix	0.80	4.8	18	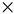	0.50	Locomotion
2015	[28]	Phoenix 3DOF	0.37	1.3	18	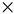	0.25	Locomotion
2015	[29]	HexaBull-1	0.53	3.4	18	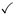	-	Locomotion
2015	[30,31]	MantisBot	0.74	6.1	28	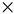	-	Navigation
2015	[32]	Snake Monster	0.70	4.6	18	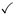	-	Locomotion
2015	[33]	BionicANT	0.15	0.105	18	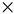	-	Swarming
2014	[34,35,36]	HECTOR	0.95	13	18	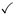	-	Navigation
2014	[37,38]	Messor II	0.30	2.5	18	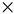	0.09	Locomotion
2014	[39,40]	LAURON V	0.90	42	24	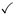	-	Navigation
2014	[41]	CREX	1	27	24	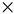	0.17	Locomotion
2012	[42]	Octavio	1	10.8	18	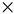	-	Locomotion
2011	[43]	-	0.46	3	18	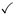	0.03	Navigation
2011	[44,45]	EduBot	0.36	3.3	6	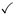	2.50	Locomotion
2010	[46]	X-RHex	0.57	9.5	6	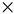	1.54	Locomotion
2008	[47]	DLR-crawler	0.50	3.5	18	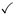	0.20	Locomotion
2006	[48,49]	AMOS-WD06	0.40	4.2	19	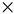	0.07	Locomotion
2006	[50,51]	Gregor I	0.30	1.2	12	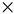	0.03	Locomotion
2005	[52,53]	BILL-Ant-a	0.33	2.3	18	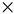	0.03	Locomotion
2001	[54]	RHex	0.54	7	6	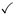	0.55	Locomotion

**Table 2 sensors-21-07609-t002:** Level of performance in terms of cost of transport (CoT) in hexapod robots walking in tripod locomotion over a flat terrain. The speed is the maximum speed.

Robot	Actuators	#Actuators	Mass (kg)	Speed (m/s)	CoT
Daisy	X-series	18	21	0.13	3.7
	X8-9 and X8-16				
HAntR	Dynamixel AX-12A	24	2.9	0.43	1.5
AntBot	Dynamixel AX-18A	18	2.3	0.90	6.2
CRABOT	Dynamixel AX-18A	24	2.5	0.05	-
Hexabot	Dynamixel XL-320	18	0.93	0.35	-
Weaver	Dynamixel	30	7	0.16	1.5–1.8
	MX-64 and MX-106				
EduBot	DC motor	6	3.3	2.5	0.5–1.6
Messor II	Dynamical RX-28	18	2.5	0.09	-
BionicANT	Trimorphic piezo-ceramic	18	0.105	-	-

**Table 3 sensors-21-07609-t003:** State-of-the-art of the different insect functionalities and behaviors ascribed to the MBs and CX. The subscript associated with each reference indicates the content of the corresponding work: analysis of insect neural circuits and associated behavioral experiments (E), modeling of neural circuits and simulations (S), neural models applied in robotic experiments (R).

Insect Brain Areas	Functionalities	References
Mushroom bodies	Olfactory learning	[261]${}_{S}$, [262]${}_{E}$
	Attention	[263]${}_{E}$, [264]${}_{S}$
	Expectation	[265]${}_{ES}$, [266]${}_{S}$
	Sameness	[234]${}_{E}$ [267]${}_{S}$
	Sequence learning	[268]${}_{SR}$ [269]${}_{SR}$
	Navigation and visual sequential memory	[260]${}_{SR}$
	Classification and decision making	[270]${}_{S}$, [271]${}_{S}$
Central complex	Navigation and detour paradigm	[272]${}_{E}$, [273]${}_{S}$, [274]${}_{S}$
	Goal-directed navigation	[275]${}_{E}$ [276]${}_{S}$
	Spatial working memory in a water maze scenario	[235]${}_{E}$
	Body-size model	[277]${}_{E}$, [35]${}_{S}$, [278]${}_{SR}$, [279]${}_{S}$
Mixed	Navigation and landmark targeting behavior	[280]${}_{S}$, [192]${}_{S}$
	Motor-skill learning	[231]${}_{E}$, [281]${}_{E}$, [282]${}_{S}$
	MBs and CX contribution to aversive visual learning	[283]${}_{E}$

## Data Availability

Not applicable.

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
