# Peer review of "Insect-Inspired Robots: Bridging Biological and Artificial Systems"

_sensors, 2021, doi:10.3390/s21227609_

Round 1
Reviewer 1 Report
The paper is a very interesting review and address design issues and research challenges in the area of hexapod robotics. The work provide historical context for a field while offering opinions on its future directions over the next years.
The title is appropriate; the abstract is focused on literature survey and conclusions. The introduction in Section 1 clearly defines the main aspects of the topic being investigated and explains the aim of the study.
Section 2 appropriately discusses the findings in biomechanics hexapod robotics according to current knowledge and existing literature. Future directions are supported by biological explanation, and study limitations are clearly highlighted.
Minor points to consider in subsequent versions:
- in table 1 of section 2 - would be interesting to introduce a reference [a] about hexapod robot with anthropomorphic architecture composed of hybrid mammalians legs with omni-wheels as feet at its extremity.
[a] Tedeschi, F.; Carbone, G. Design of a Novel Leg-Wheel Hexapod Walking Robot. Robotics 2017, 6, 40. https://doi.org/10.3390/robotics6040040
- according to the topics of Sensors journal, section 2.1.4 in my opinion should be extended illustrating more in-depth the achievements in the field and providing generalized background of the subject, for example, highlighting advantages and drawbacks of varius solution
- in section 2.2.2 - i suggest the authors to introduce a reference [b], as example of kinematic changes in robot locomotion taking account a scale of robotic platform and mechanics material
[b] Hydraulically Actuated Hexapod Robots, Nonami, K., Barai, R.K., Irawan, A., Daud, M.R., Springer 2014, ISBN 9784431543497
Section 3 address locomotion control, involving interlimb coordination, intralimb coordination and joint compliance. The section present also different robot locomotion control methods.
Minor points to consider in subsequent versions:
- in my opinion the quality of paper should be improuved with a short section or a picture illustrating more in depth hexapod walking gaits. I feel this is an important backgound helpful for the readers, and therefore it merits more discussion.
Section 4 exhaustively describes and discuss high level cognition control. The section is well researched and nicely written.
The reference does not include inappropriate self-citations.
Conclusion:
After minor revision the paper shoud be a worthwhile resource for scientists.
Reviewer 2 Report
The article “Insect-inspired Robots: Bridging Biological and Artificial Systems” by Manoonpong et al. reviews several contributions focused on hexapod robots, addressing a number of questions in this context.
The article is quite interesting but several issue needs to be addressed before having an article suitable for acceptance.
The article seems to collect a number of previous research related to locomotion, control, etc. In this way, the reader has a sort of list of contributions but it is needed something linking them in a big vision. Maybe a figure that indicate how these robot evolved and what are the main challenges would help to create a “story” in the review.
Concerning the bioinspired control, authors mainly refers to few insect models (Drosophila).
However, there are many recent works that report significant discoveries related to insects brains and cognition, including lateralization, social learning, sensory-motor compensatory strategies based on cognitive processes. These contribution have been found in fly, beetles, locusts, etc…
I suggest authors to do a further effort and expand this part, adding relevant researches from literature.
Some suggestions to cite and comment are
Romano, D., Benelli, G., Kavallieratos, N. G., Athanassiou, C. G., Canale, A., & Stefanini, C. (2020). Beetle-robot hybrid interaction: sex, lateralization and mating experience modulate behavioural responses to robotic cues in the larger grain borer Prostephanus truncatus (Horn). Biological Cybernetics, 114(4), 473-483.
Bell, A. T., & Niven, J. E. (2016). Strength of forelimb lateralization predicts motor errors in an insect. Biology letters, 12(9), 20160547.
But many additional works can be found by authors.
The review lack of a part where authors comment on the lesson learned from their review.
A deep English revision is needed.
Round 2
Reviewer 2 Report
Authors addressed almost all my comments and now the manuscript is much improved. I saw Authors mentioned also social learning in insect.
This part could be improved referring to social learnign as high-level complex cognitive process also reported in non-social insiects.
Some useful work that may help Authors in improve the scientific value of their work are:
Romano, D., Benelli, G., & Stefanini, C. (2021). Opposite valence social information provided by bio-robotic demonstrators shapes selection processes in the green bottle fly. Journal of the Royal Society Interface, 18(176), 20210056. https://doi.org/10.1098/rsif.2021.0056
Sarin, S., & Dukas, R. (2009). Social learning about egg-laying substrates in fruitflies. Proceedings of the Royal Society B: Biological Sciences, 276(1677), 4323-4328.
Coolen, I., Dangles, O., & Casas, J. (2005). Social learning in noncolonial insects?. Current biology, 15(21), 1931-1935.
Author Response
For this second round, our added improvements to the article will be written in red.
Thank you for your recommendation.